# FlowSymm: Physics Aware, Symmetry Preserving Graph Attention for Network Flow Completion

**Ege Demirci**
Department of Computer Science, UC Santa Barbara
Santa Barbara, CA 93106, USA

**Francesco Bullo**
Center for Control, Dynamical Systems, and Computation, UC Santa Barbara
Santa Barbara, CA 93106, USA

**Ananthram Swami**
U.S. Army DEVCOM Research Laboratory
Adelphi, MD 20783, USA

**Ambuj Singh**
Department of Computer Science, UC Santa Barbara
Santa Barbara, CA 93106, USA

## Abstract

Recovering missing flows on the edges of a network, while exactly respecting local conservation laws, is a fundamental inverse problem that arises in many systems such as transportation, energy, and mobility. We introduce FlowSymm, a novel architecture that combines (i) a *group-action* on divergence-free flows, (ii) a graph-attention encoder to learn feature-conditioned weights over these symmetry-preserving actions, and (iii) a lightweight Tikhonov refinement solved via implicit bilevel optimization. The method first anchors the given observation on a minimum-norm divergence-free completion. We then compute an orthonormal basis for all admissible group actions that leave the observed flows invariant and parameterize the valid solution subspace, which shows an Abelian group structure under vector addition. A stack of GATv2 layers then encodes the graph and its edge features into per-edge embeddings, which are pooled over the missing edges and produce per-basis attention weights. This attention-guided process selects a set of physics-aware group actions that preserve the observed flows. Finally, a scalar Tikhonov penalty refines the missing entries via a convex least-squares solver, with gradients propagated implicitly through Cholesky factorization. Across three real-world flow benchmarks (traffic, power, bike), FlowSymm outperforms state-of-the-art baselines in RMSE, MAE and correlation metrics.

## 1 Introduction

Estimating missing flows in a network is a fundamental inverse problem with significant applications in transportation planning, grid reliability, and urban mobility. In practice, only a fraction of the edges are equipped with sensors, and simply imputing the rest can result in physically implausible predictions. For example, this could lead to traffic counts that violate conservation laws at intersections, power flows that overlook Kirchhoff's laws, or bike flows that disregard bike stations' capacities. Algebraically, these laws are captured by the node-balance equation $Bf = c$, where $B$ is the incidence matrix, $f$ the edge-flow vector, and $c$ unknown net injections. Real systems demonstrate approximate conservation rather than perfect balance due to factors such as noise, unobserved sources

and sinks, and measurement errors. Therefore, a successful flow estimator must learn from the data while respecting the underlying physical principles.

## 1.1 RELATED WORK

Recent advances in *physics-informed machine learning* incorporate conservation laws via *soft* penalties or bilevel-learned regularizers (Gao et al., 2022; Seccamonte et al., 2023; Sharma & Fink, 2025; Usama et al., 2022; Wang et al., 2022; Zhang et al., 2024). For example, Silva et al. (2021) learn edge-specific regularization weights that trade-off flow conservation against data fidelity, yielding significant accuracy gains on traffic and power benchmarks. Other works, such as Mi et al. (2025), embed conservation directly into message-passing rules, splitting flux exchanges into symmetric and asymmetric components to ensure implicit continuity without over-constraining the model. These approaches balance realism (allowing some divergence) and physical consistency.

Parallel research thrusts have shown that encoding *symmetries* via *group-equivariant* Graph Neural Networks (GNNs) markedly improves sample efficiency and generalization in physical domains (Bronstein et al., 2021; Finzi et al., 2021; Pearce-Crump & Knottenbelt, 2024). Architectures such as the $E(n)$ Equivariant Graph Network (EGNN) guarantee that predictions commute with Euclidean transformations, honoring rotational and translational invariances inherent in road grids and power meshes (Satorras et al., 2022). Hierarchical and gauge-equivariant extensions further capture multi-scale or manifold symmetries and demonstrate state-of-the-art performance on molecular and fluid simulations (Han et al., 2022; Toshev et al., 2023). Beyond these, Murphy et al. (2019) introduced relational pooling, which aggregates representations over the permutation group to further boost discrimination power. In continuous domains, Tensor Field Networks (Thomas et al., 2018), SE(3)-Transformer (Fuchs et al., 2020), and NequIP (Batzner et al., 2022) enforce E(3)-equivariance via spherical harmonics and tensor representations, demonstrating significant gains in molecular and physical prediction tasks. However, these methods primarily address *spatial* or *geometric* symmetries (invariance to rotation/translation). In contrast, flow completion requires respecting *algebraic* symmetries defined by the linear conservation constraints and sensor masks, rather than the geometric embedding of the graph.

A third line of research integrates *implicit differentiation* and *bilevel optimization* to learn hyperparameters, such as regularization strengths and group-action coefficients, end to end. Classical implicit-differentiation techniques allow gradients to flow through optimization layers without unrolling every iteration, greatly improving scalability. Gradient-based bilevel schemes have powered hyperparameter and meta-learning breakthroughs in deep networks (Domke, 2012; Franceschi et al., 2017; 2018), and recent work shows how to incorporate solver structures (e.g., implicit time-stepping, finite-volume flux updates) directly into GNN architectures for stable long-horizon prediction (Horie & Mitsume, 2022; 2024). Recent works such as Smith et al. (2022) and Gu et al. (2020) embed implicit physics-informed solvers or fixed-point equilibrium layers within GNN architectures, leveraging implicit differentiation to backpropagate through mass-conservation and equilibrium constraints end-to-end. Likewise, Golias et al. (2024) and Hao et al. (2023) integrate combinatorial (successive-shortest-path) and PDE-constrained solvers into bilevel optimization frameworks, differentiating through these solvers to jointly learn solver parameters and network weights with improved stability and efficiency. Finally, we note the growing interest in operator learning for continuous physics, such as Neural Operators (Lu et al., 2022) and Galerkin Transformers (Cao, 2021). While these models are powerful for solving PDEs on continuous domains, adapting them to discrete, irregular graphs with partial observability remains a distinct challenge and a complementary direction for future work.

## 1.2 OUR APPROACH

We introduce FLOWSYMM, a solver that augments a graph attention backbone with symmetry-preserving, physics-aware corrections tailored to partially observed flow graphs. Our formulation is guided by four complementary design principles. **(i)** First, we reinterpret admissible divergence-free adjustments as elements of an Abelian *group action*; by constructing an orthonormal basis and retaining only the leading $k$ basis vectors, we obtain a tractable latent space that scales linearly with the number of missing flows. Unlike typical equivariant GNNs, which embed spatial or permutation symmetries, our group-action basis directly spans the algebraic space of divergence-free adjustments tailored to the missing flows. **(ii)** Second, we combine this basis with edge-wise GATv2

embeddings (Brody et al., 2022): the attention mechanism scores each basis vector in a context-aware manner, enabling the model to inject corrective flows where and when the local structure demands, while preserving flow balance on the anchor manifold. This attention-guided selection departs from prior bilevel diagonal regularizers by coupling missing edges through context-dependent weights over physically interpretable basis actions, rather than treating each edge independently. **(iii)** Third, we impose *soft* physical consistency through a feature-conditioned Tikhonov refinement and train all weights end-to-end via reverse-mode *implicit differentiation*. **(iv)** Finally, we demonstrate through extensive experiments on three real-world benchmarks —traffic, power, and bike— that FLOWSYMM consistently outperforms nine baselines. Our approach reduces RMSE by up to ten percent while yielding attention maps and regularization weights that align with domain-specific intuition. Taken together, these contributions demonstrate that coupling learned graph attention with symmetry-preserving flow corrections offers a simple and interpretable route to state-of-the-art network-flow completion.

## 2 FLOW ESTIMATION PROBLEM

We are given a directed graph $\mathcal{G} = (\mathcal{V}, \mathcal{E}, X)$ with $n = |\mathcal{V}|$ vertices and $m = |\mathcal{E}|$ edges. Every edge $e \in \mathcal{E}$ carries a $d$-dimensional feature vector $X_e \in \mathbb{R}^d$, stacked row-wise into $X \in \mathbb{R}^{m \times d}$. We frame sensor readings as noisy, partial snapshots of an underlying flow that should nearly, but not exactly, obey conservation. The goal is to reconstruct the unobserved edges while respecting these physical constraints ( specifically, flow conservation $Bf = c$. ) Let $B \in \{-1, 0, 1\}^{n \times m}$ denote the oriented incidence matrix of $\mathcal{G}$, and let $c \in \mathbb{R}^n$ be the vector of net nodal injections (with $c_i < 0$ for sources and $c_i > 0$ for sinks).

An *ideal* flow $f \in \mathbb{R}^m$ therefore satisfies the balance equation $Bf = c$, reducing to classical conservation when $c = \mathbf{0}$. Flows are observed on a subset $\mathcal{E}_{\text{obs}} \subseteq \mathcal{E}$, leaving $\mathcal{E}_{\text{miss}} = \mathcal{E} \setminus \mathcal{E}_{\text{obs}}$. We denote $m_{\text{obs}} = |\mathcal{E}_{\text{obs}}|$ and $m_{\text{miss}} = |\mathcal{E}_{\text{miss}}|$. We *define* binary selector matrices $S_{\text{obs}} \in \{0, 1\}^{m_{\text{obs}} \times m}$ and $S_{\text{miss}} \in \{0, 1\}^{m_{\text{miss}} \times m}$ and write, for any $f \in \mathbb{R}^m$,

$$f^{\text{obs}} = S_{\text{obs}}f, \quad \delta^{\text{miss}} = S_{\text{miss}}f, \quad B_{\text{obs}} = B\,S_{\text{obs}}^\top, \quad B_{\text{miss}} = B\,S_{\text{miss}}^\top.$$

We lift $\delta^{\text{miss}} \in \mathbb{R}^{m_{\text{miss}}}$ into $\mathbb{R}^m$ via $S_{\text{miss}}^\top \delta^{\text{miss}}$. A noisy observation $\hat{f} \in \mathbb{R}^m$ satisfies $\hat{f}_e = f_e + \varepsilon_e$ for $e \in \mathcal{E}_{\text{obs}}$ and $\hat{f}_e = 0$ on $\mathcal{E}_{\text{miss}}$. We make no explicit assumption on the statistical distribution of the noise; our reconstruction is agnostic to any particular noise model, and abbreviate the measured part as $\hat{f}^{\text{obs}} := S_{\text{obs}}\hat{f}$.

Our goal is to reconstruct the missing values $\{f_e : e \in \mathcal{E}_{\text{miss}}\}$. So, we learn a mapping $(\mathcal{G}, \hat{f}, c) \mapsto \tilde{f}_\theta \in \mathbb{R}^m$ that (i) fits the noisy measurements (small $\|S_{\text{obs}}\tilde{f}_\theta - \hat{f}^{\text{obs}}\|_2$), (ii) keeps the balance residual small (small $\|B\tilde{f}_\theta - c\|_2$), (iii) operates within (and respects) the admissible divergence-free group-action subspace (i.e., the null-space of valid adjustments) that leaves observed flows unchanged, and (iv) couples missing edges through features $X$ rather than a diagonal penalty.

Earlier approaches enforced *hard* conservation $Bf = \mathbf{0}$ (Jia et al., 2019), used a *diagonal* feature prior (Silva et al., 2021), or ignored the symmetry altogether. This matters because in flow completion, not modifying sensors is a hard operational constraint; our approach guarantees this while still exploring a rich (truncated) null-space of divergence-free *redistributions*. This is a different symmetry from Euclidean or permutation equivariances commonly studied in equivariant GNNs (Bronstein et al., 2021; Finzi et al., 2021; Pearce-Crump & Knottenbelt, 2024); it is an algebraic symmetry determined by the incidence matrix together with the sensor mask (the partition $\mathcal{E} = \mathcal{E}_{\text{obs}} \cup \mathcal{E}_{\text{miss}}$), which freezes observed edges and allows adjustments only on missing ones.

The initial estimate respects conservation (i.e., it satisfies $Bf = c$, reducing to divergence-free only when $c = \mathbf{0}$). The final prediction is then allowed a small, learnable violation of node balance so it can handle sensor noise and better fit the measurements, incurring only a small residual relative to $c$. We provide a list of symbols in Appendix A.

# 3 OUR APPROACH: FLOWSYMM

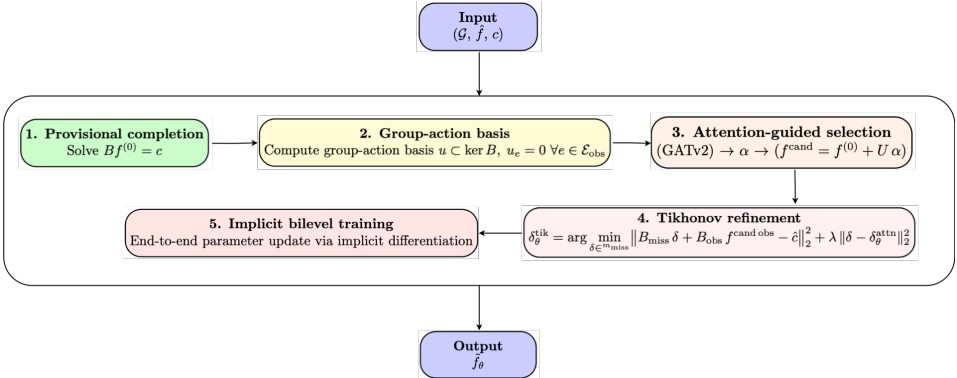

Figure 1: **Pipeline of FLOWSYMM**: (1) construct a provisional balanced completion, (2) build a divergence-free group-action basis, (3) score and combine these actions via attention, (4) refine the missing flows with a feature-conditioned Tikhonov solver, and (5) train all parameters end-to-end via implicit bilevel optimization.

## 3.1 INITIAL BALANCED COMPLETION

We start by anchoring the partially observed snapshot to the *closest* balanced flow so that all later updates stay within the physical manifold. Given the partially observed flow vector $\hat{f} \in \mathbb{R}^m$ (zeros on all missing flows) and the injection vector $c \in \mathbb{R}^n$ summarizing known sources ($c_i < 0$) and sinks ($c_i > 0$), our first goal is to build a *provisional* full flow

$$f^{(0)} = \hat{f} + S_{\mathrm{miss}}^\top \delta^{(0)}, \qquad B_{\mathrm{miss}} \delta^{(0)} = c - B_{\mathrm{obs}} \hat{f}^{\mathrm{obs}}, \qquad \hat{f}^{\mathrm{obs}} := S_{\mathrm{obs}} \hat{f}.$$

We choose the *minimum-norm* solution via the Moore–Penrose pseudoinverse,

$$\delta^{(0)} = B_{\mathrm{miss}}^\top \left( B_{\mathrm{miss}} B_{\mathrm{miss}}^\top \right)^\dagger \left( c - B_{\mathrm{obs}} \hat{f}^{\mathrm{obs}} \right).$$

Here $(\cdot)^\dagger$ denotes the Moore–Penrose pseudoinverse; when $B_{\mathrm{miss}} B_{\mathrm{miss}}^\top$ is invertible the dagger reduces to the usual matrix inverse. In implementation we compute $\delta^{(0)}$ with a numerically stable least-squares routine (QR or SVD) to handle possible rank deficiency. We pick the minimum-norm solution among all $\delta$ satisfying $B_{\mathrm{miss}} \delta = c - B_{\mathrm{obs}} \hat{f}^{\mathrm{obs}}$ to avoid introducing unnecessarily large adjustments on the missing edges; by construction this guarantees $B f^{(0)} = c$ while leaving every observed entry intact. Although the choice of $f^{(0)}$ is not unique, it provides a convenient *anchor* in $\mathcal{F}_{\mathrm{bal}}(c) := \{ f \in \mathbb{R}^m : B f = c \}$.

Prior work either fixes $c = \mathbf{0}$ or enforces balance with a global diagonal penalty; our anchor keeps the *exact* injection profile $c$ while requiring only a sparse linear solve, providing a principled starting point for learned corrections. In other words, because $f^{(0)}$ already realizes the prescribed injections $c$, all subsequent *valid adjustments* of interest are taken from the subspace $\mathcal{A} = \{ u \in \ker B : u_e = 0 \ \forall e \in \mathcal{E}_{\mathrm{obs}} \}$, so that updates $f^{(0)} + u$ remain on the same affine manifold and continue to leave observed entries unchanged. Hence the overall method preserves the conservation pattern defined by $c$ by design.

## 3.2 GROUP–ACTION BASIS

Intuitively, a group action is any redistribution of the current flow that (i) maintains the nodal balance and (ii) leaves every instrumented edge untouched. First, observe that the set of all group actions

$$\mathcal{A} = \left\{ u \in \mathbb{R}^m : B u = 0, \ u_e = 0 \ \forall e \in \mathcal{E}_{\mathrm{obs}} \right\}$$

is an *abelian group* under vector addition. This group acts on any balanced flow $f \in \mathcal{F}_{\mathrm{bal}}(c) = \{ f : B f = c \}$:

$$g_\alpha : f \mapsto f + U \alpha, \qquad \alpha \in \mathbb{R}^k,$$

where $U \in \mathbb{R}^{m \times k}$ is a column-orthonormal basis for $\mathcal{A}$. The identity corresponds to $\alpha = 0$; composition satisfies $g_\alpha \circ g_\beta = g_{\alpha+\beta}$; and each element is invertible via $g_\alpha^{-1} = g_{-\alpha}$. Because every column of $U$ lies in $\ker B$ and vanishes on $\mathcal{E}_{\mathrm{obs}}$, the action preserves both the prescribed injection and the observed flows. In order to construct $U$, we note that

$$\mathcal{A} = \ker B \cap \{u : u_e = 0 \ \forall e \in \mathcal{E}_{\mathrm{obs}}\},$$

whose dimension is $r = m_{\mathrm{miss}} - \mathrm{rank}(B_{\mathrm{miss}}) \geq m_{\mathrm{miss}} - (n-1)$ for a connected graph (with equality when $B_{\mathrm{miss}}$ has row rank $n-1$). In practice we *truncate* this space and keep only the first $k$ basis vectors to control capacity and compute time; in our experiments, we found that retaining $k = 256$ basis vectors was sufficient to capture these variations (see Sec. 4.3). Formally, let

$$P_{\mathrm{bal}} = I_m - B^\dagger B, \qquad (\text{orthogonal projector onto } \ker B),$$

and define the projector onto $\mathcal{A}$ by

$$P_{\mathcal{A}} = P_{\mathrm{bal}} - P_{\mathrm{bal}} S_{\mathrm{obs}}^\top (S_{\mathrm{obs}} P_{\mathrm{bal}} S_{\mathrm{obs}}^\top)^\dagger S_{\mathrm{obs}} P_{\mathrm{bal}}.$$

Intuitively, this formula first selects the space of all divergence-free flows (using $P_{\mathrm{bal}}$) and then subtracts the specific components that would violate the sensor-invariance constraint. We then take $U$ as the top $k$ singular vectors from the Singular Value Decomposition (SVD) of $\mathrm{range}(P_{\mathcal{A}}) \subset \mathbb{R}^m$, which guarantees $BU = 0$ and $S_{\mathrm{obs}} U = 0$.

Unlike Euclidean-equivariant GNNs that encode rotations or permutations, our basis spans *algebraic* symmetries, zero-divergence moves specific to the observed/missing split, providing a customized latent space for flow completion. Note that our model learns weights for these specific, ordered physical modes, and is therefore not invariant to arbitrary rotations of the basis $U$. Any linear combination $U\alpha$ therefore belongs to $\mathcal{A}$; adding $U\alpha$ to the anchor $f^{(0)}$ moves strictly within the divergence-free affine subspace while leaving all observed edges untouched.

### 3.3 ATTENTION-GUIDED GROUP-ACTION SELECTION

Having enumerated a physics-valid basis, we now *learn* which directions matter for the current graph snapshot. Let $L \in \mathbb{N}$ be the number of stacked GATv2 layers. We apply $L = 2$ stacked GATv2 layers to the edge features $X$, producing

$$H = \mathrm{GATv2}_\theta(\mathcal{G}, X) \in \mathbb{R}^{m \times d'}, \qquad H = \begin{bmatrix} H_1^\top \\ \cdots \\ H_m^\top \end{bmatrix}.$$

Here $H_e \in \mathbb{R}^{d'}$ denotes the embedding of edge $e$ and $d'$ is the hidden dimension.

For every missing flow $e \in \mathcal{E}_{\mathrm{miss}}$ and basis index $i \in \{1, \ldots, k\}$ we compute a *compatibility score* (effectively a local 'vote'); the factor $|u_e^{(i)}|$ modulates the action's strength on edge $e$ and down-weights bases that do not touch $e$:

$$q_{e,i} = (w_i^\top H_e) |u_e^{(i)}|, \qquad w_i \in \mathbb{R}^{d'} \text{ trainable.}$$

In matrix form $Q = [q_{e,i}] \in \mathbb{R}^{m_{\mathrm{miss}} \times k}$. Aggregating over missing edges and applying a softmax yields

$$s_i = \frac{1}{m_{\mathrm{miss}}} \sum_{e \in \mathcal{E}_{\mathrm{miss}}} q_{e,i}, \qquad s \in \mathbb{R}^k,$$

$$\alpha_\theta = \mathrm{softmax}(s), \qquad \sum_{i=1}^{k} \alpha_{\theta,i} = 1, \ \alpha_{\theta,i} > 0.$$

The resulting action is $\Delta = U\alpha_\theta = \sum_{i=1}^{k} \alpha_{\theta,i} u^{(i)}$, and the *candidate* balanced flow becomes $f_\theta^{\mathrm{cand}} = f^{(0)} + \Delta$. We write $f_\theta^{\mathrm{cand,obs}} := S_{\mathrm{obs}} f_\theta^{\mathrm{cand}}$. Since each $u^{(i)} \in \ker B$ and vanishes on

$\mathcal{E}_{\text{obs}}$, we retain $Bf_\theta^{\text{cand}} = c$ and $S_{\text{obs}}f_\theta^{\text{cand}} = \hat{f}^{\text{obs}}$. Finally, we define the candidate missing vector $\delta_\theta^{\text{attn}} := S_{\text{miss}}f_\theta^{\text{cand}} \in \mathbb{R}^{m_{\text{miss}}}$.

The term $q_{e,i}$ measures how much the local context of edge $e$ votes for the utilization of basis vector $i$. Each basis vector represents a global adjustment that spans multiple edges. Intuitively, the weight on this basis vector is the sum of the attention scores on its constituent edges. Summing these scores across all missing flows yields a global vector $s \in \mathbb{R}^k$, ensuring that the final global correction $\Delta$ is determined by the aggregate demands of local features. Applying a softmax turns $s$ into a probability distribution $\alpha_\theta$ over the $k$ basis vectors. Figure 2 shows the histogram of these learned attention weights $\{\alpha_{\theta,i}\}_{i=1}^k$ for the Traffic, Power, and Bike datasets (see Sec. 4). Although every basis vector receives some weight, most of the mass concentrates in a few of them, indicating that the model focuses on a small subset of group actions.

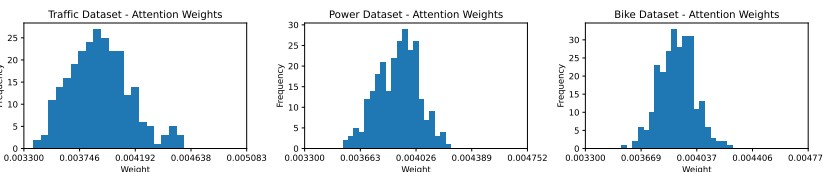

Figure 2: Distribution of learned attention weights $\alpha_{\theta,i}$ over $k=256$ basis actions on Traffic, Power, and Bike.

### 3.4 FEATURE-CONDITIONED TIKHONOV REFINEMENT

The first three steps produce a balanced candidate $f_\theta^{\text{cand}}$. This step allows a small, controlled deviation on the *missing* edges to absorb noise in the observations. The attention step outputs $f_\theta^{\text{cand}} = f^{(0)} + \Delta$ (with $Bf_\theta^{\text{cand}} = c$ and $S_{\text{obs}}f_\theta^{\text{cand}} = \hat{f}^{\text{obs}}$). We employ Tikhonov regularized least-squares to cope with measurement noise and encoder uncertainty.

Even though the observations are noisy, they imply an empirical imbalance $\hat{c} := B_{\text{obs}}\hat{f}^{\text{obs}} \in \mathbb{R}^n$, which may differ slightly from the prescribed $c$. We use $\hat{c}$ only to measure the empirical residual implied by noisy sensors; $c$ remains the physics target. The refinement moves *toward* reduced residual relative to $\hat{c}$ while staying close to the candidate on the missing edges, $\delta_\theta^{\text{attn}} := S_{\text{miss}}f_\theta^{\text{cand}} \in \mathbb{R}^{m_{\text{miss}}}$.

$$\delta_\theta^{\text{tik}} = \arg\min_{\delta \in \mathbb{R}^{m_{\text{miss}}}} \underbrace{\|B_{\text{miss}}\delta + B_{\text{obs}}f_\theta^{\text{cand obs}} - \hat{c}\|_2^2}_{\text{divergence w.r.t. empirical imbalance}} + \underbrace{\lambda\|\delta - \delta_\theta^{\text{attn}}\|_2^2}_{\text{Tikhonov pull toward the candidate}}. \tag{1}$$

The strictly convex objective yields the normal equations

$$\left(B_{\text{miss}}^\top B_{\text{miss}} + \lambda I_{m_{\text{miss}}}\right)\delta_\theta^{\text{tik}} = \lambda\,\delta_\theta^{\text{attn}} - B_{\text{miss}}^\top\left(B_{\text{obs}}f_\theta^{\text{cand,obs}} - \hat{c}\right),$$

and the closed form

$$\delta_\theta^{\text{tik}} = \left(B_{\text{miss}}^\top B_{\text{miss}} + \lambda I_{m_{\text{miss}}}\right)^{-1}\left(\lambda\,\delta_\theta^{\text{attn}} - B_{\text{miss}}^\top\left(B_{\text{obs}}f_\theta^{\text{cand,obs}} - \hat{c}\right)\right).$$

This provides an inexpensive shift towards a lower residual divergence. Updating only the missing components, we get $\tilde{f}_\theta = f_\theta^{\text{cand}} + S_{\text{miss}}^\top\left(\delta_\theta^{\text{tik}} - \delta_\theta^{\text{attn}}\right)$. Because $B_{\text{miss}}^\top B_{\text{miss}} + \lambda I_{m_{\text{miss}}}$ is *symmetric positive definite* (SPD), we use Cholesky (or CG) and back-propagate exact gradients via implicit differentiation. In short, the refinement is a single SPD solve that trades reduced empirical divergence for staying close to the attention-guided candidate on the missing edges.

### 3.5 IMPLICIT BILEVEL TRAINING

We tune all parameters by minimizing a held-out edge error while back-propagating exactly through the Tikhonov solver, rather than unrolling its iterations, which produces exact hyper-gradients at minimal memory cost. All learnable parameters $\theta = (\text{GATv2 weights}, \lambda)$ are trained end-to-end by minimizing a held-out loss while *implicitly differentiating* through the inner solver.

Let $\{(\mathcal{E}_{\text{obs}}^{(k)}, \hat{f}^{(k)}, g^{\text{true},(k)})\}_{k=1}^{K}$ be a $K$-fold split: fold $k$ provides observed edges $\mathcal{E}_{\text{obs}}^{(k)}$, a partially masked input $\hat{f}^{(k)} \in \mathbb{R}^m$ (zeros on $\mathcal{E}_{\text{miss}}^{(k)}$), and ground-truth flows $g^{\text{true},(k)} \in \mathbb{R}^{m_{\text{val}}^{(k)}}$ on a validation set $\mathcal{E}_{\text{val}}^{(k)} \subseteq \mathcal{E}_{\text{miss}}^{(k)}$. Let $S_{\text{val}}^{(k)} \in \{0,1\}^{m_{\text{val}}^{(k)} \times m_{\text{miss}}^{(k)}}$ extract those coordinates.

Running the first three steps on fold $k$ yields $f_\theta^{\text{cand},(k)}$ and $\delta_\theta^{\text{attn},(k)} := S_{\text{miss}}^{(k)} f_\theta^{\text{cand},(k)}$.

*Inner solver.* On each fold, we solve the Tikhonov problem in equation 1 with fold-indexed quantities, producing $\delta_\theta^{\text{tik},(k)}$. The associated normal equations mirror §3.4 and are omitted for brevity.

*Outer loss.* The validation loss averages squared error on held-out edges:

$$\mathcal{L}_{\text{val}}(\theta) = \frac{1}{K} \sum_{k=1}^{K} \big\| S_{\text{val}}^{(k)} \big( \delta_\theta^{\text{tik},(k)} - g^{\text{true},(k)} \big) \big\|_2^2.$$

*Implicit gradients.* To compute $\nabla_\theta \mathcal{L}_{\text{val}}$ without unrolling, we use reverse-mode implicit differentiation. For each fold $k$, solve the adjoint system

$$\big( B_{\text{miss}}^{(k)\top} B_{\text{miss}}^{(k)} + \lambda I_{m_{\text{miss}}^{(k)}} \big) w^{(k)} = S_{\text{val}}^{(k)\top} \big( S_{\text{val}}^{(k)} \delta_\theta^{\text{tik},(k)} - g^{\text{true},(k)} \big),$$

and then back-propagate through the full right-hand side

$$b^{(k)} = \lambda \, \delta_\theta^{\text{attn},(k)} \; - \; (B_{\text{miss}}^{(k)})^\top \Big( B_{\text{obs}}^{(k)} f_\theta^{\text{cand,obs},(k)} - \hat{c}^{(k)} \Big),$$

i.e., account for the dependence of $\lambda$, $\delta_\theta^{\text{attn},(k)}$, and $f_\theta^{\text{cand,obs},(k)}$ on $\theta$. Accumulating over folds yields $\nabla_\theta \mathcal{L}_{\text{val}}$. Each outer update thus requires exactly two SPD solves per fold for the inner layer (one forward to obtain $\delta^{\text{tik}}$, one adjoint to obtain $w$) and yields *exact* hyper-gradients, helped by the strong convexity of the inner problem and the divergence-free constraints imposed earlier. Additional code-level details are in Appendix E.

## 4 EXPERIMENTS

We evaluate FLOWSYMM on three real-world flow graphs that span transportation, energy and micromobility domains. For every dataset we (i) keep the original directed/undirected topology, (ii) normalize flow magnitudes to $[0, 1]$, (iii) encode the features in a one-hot manner, and (iv) follow a 10-fold edge hold-out protocol: in each fold a disjoint subset $\mathcal{E}_{\text{obs}}$ is revealed for training, while the complementary $\mathcal{E}_{\text{val}}$ is used for validation/testing. For a more detailed description of the datasets and results, please refer to the Appendix C.

Real transportation and power networks are much larger, but *reliable, edge-level ground truth flows* are typically available only on *sensor-instrumented, clustered subgraphs* comprising a few thousand links. Acquiring dense ground truth at city, or country-scale remains prohibitively expensive or impossible. These clustered subsets reflect the partial-observability scenario that our method addresses. To enable apples-to-apples comparison, we *reuse the Traffic and Power configurations from* Silva et al. (2021) (same sources and preparation protocol). In addition, we *introduce a new Bike micromobility dataset* built from Citi Bike trip logs (Mar 2025), which broadens the evaluation to an undirected domain. Although public datasets are sparse, we provide a complexity analysis in Appendix D.

**Traffic:** LA County road network as a directed graph (junctions as nodes, road segments as edges) built from OpenStreetMap and compressed by merging non-branching chains; $n = 5{,}749$, $m = 7{,}498$ (OpenStreetMap contributors, 2017; Silva et al., 2021). Daily average counts from Caltrans PeMS (2018) are matched to edges, yielding measurements on 2,879 edges (38%); each edge has 18 features (California Department of Transportation, 2018). Measurements are moderately noisy due to sensor issues and matching errors.

**Power:** Continental Europe transmission grid as an undirected graph from PyPSA-Eur: $n = 2{,}048$ buses, $m = 2{,}729$ lines/connectors with active power flows (MW) from a single snapshot; 14 edge features (Brown et al., 2018; Hörsch et al., 2018).

**Bike:** Micromobility graph from Citi Bike (Mar 2025): $n = 1{,}966$ active stations; undirected edges for the 2,647 most-frequent station pairs using bidirectional trip totals; flows are average daily transfers; 9 edge features (Citi Bike NYC, 2025).

## 4.1 EVALUATION PROTOCOL AND BASELINES

Performance is quantified with two complementary statistics computed on the held-out edges: the root–mean–squared error (RMSE), mean–absolute error (MAE), and the Pearson correlation coefficient (CORR). Formal definitions of these metrics are in Appendix B. All results in Table 1 are the mean across the ten test splits.

We compare FLOWSYMM against nine baselines spanning purely physics-based, purely data-driven, hybrid and bilevel formulations. Div enforces conservation by minimizing the node-wise divergence with a single regularization parameter $\lambda$, which we select via line search on a held-out validation set (Jia et al., 2019). MLP is a two-layer feedforward network with ReLU activations in both layers, trained to predict edge flows solely from their 18/14/9-dimensional feature vectors. GCN is a two-layer graph convolutional model using Chebyshev polynomials of degree 2 (Defferrard et al., 2017; Kipf & Welling, 2017) and ReLU non-linearities, which incorporates both edge attributes and graph topology but does not impose any conservation constraint. To strengthen the data-driven baselines with recent expressive architectures, we additionally include: (i) **EGNN** (Satorras et al., 2022) (two EGNNConv layers), which is $E(n)$-equivariant to Euclidean transformations; we apply it at the *edge level* via the line-graph construction with per-edge features; and (ii) **GIN** (Xu et al., 2018) (two GINConv layers with sum aggregator), which is maximally powerful under the Weisfeiler–Lehman test. Both use the same tuning protocol as GCN/MLP (learning rates $\{10^{-1}, 10^{-2}, 10^{-3}\}$, widths $\{8, 16, 32\}$, early stopping on validation). We also consider two hybrid variants: MLP-Div applies the MLP's predictions as a prior to Div, and GCN-Div does the same with the GCN outputs; in both hybrids the regularization weight $\lambda$ is again tuned on the validation split. We evaluated the bilevel diagonal regularizer framework of Silva et al. (2021), the previous state-of-the-art, instantiated with an MLP encoder (Bil-MLP) and a GCN encoder (Bil-GCN).

## 4.2 RESULTS

Table 1 summarizes the performance of FLOWSYMM against nine baselines, emphasizing the comparison to Bil-GCN, the prior state-of-the-art. Across all evaluated metrics, FLOWSYMM consistently achieves the best results, demonstrating the effectiveness of integrating physics-aware, symmetry-preserving group actions with attention-based encodings and feature-conditioned refinements.

Specifically, FLOWSYMM shows substantial improvements over the previous state-of-the-art baseline, Bil-GCN, reducing RMSE by approximately 8% (Traffic), 10% (Power), and 9% (Bike). Similarly, MAE improvements over Bil-GCN are notable, with reductions of approximately 16% (Traffic), 9% (Power), and 12% (Bike). These results clearly demonstrate the additional predictive power provided by explicitly incorporating group actions rather than relying solely on diagonal regularizers learned via bilevel optimization. Additionally, FLOWSYMM achieves higher Pearson correlation (CORR) values compared to all baselines, indicating that predictions better capture the true underlying flow patterns. This advantage becomes particularly meaningful in real-world scenarios where understanding flow trends is essential for reliable operational decisions.

Table 1: **Comparison of prediction accuracy across baselines on the Traffic, Power, and Bike datasets.** FLOWSYMM achieves the lowest RMSE, MAE, and the highest Pearson correlation (CORR) in all three domains. Relative to the previous state-of-the-art Bil-GCN, FlowSymm reduces RMSE by approximately 8% (Traffic), 10% (Power), and 9% (Bike), while also lowering MAE by up to 16% and improving correlation by up to 0.04.

|  | Traffic | | | Power | | | Bike | | |
|---|---|---|---|---|---|---|---|---|---|
| Method | RMSE | MAE | CORR | RMSE | MAE | CORR | RMSE | MAE | CORR |
| Div | $0.071 \pm 0.007$ | $0.041 \pm 0.009$ | 0.76 | $0.036 \pm 0.006$ | $0.018 \pm 0.003$ | 0.92 | $0.039 \pm 0.007$ | $0.018 \pm 0.003$ | 0.91 |
| MLP | $0.083 \pm 0.005$ | $0.055 \pm 0.006$ | – | $0.069 \pm 0.008$ | $0.042 \pm 0.007$ | 0.56 | $0.073 \pm 0.008$ | $0.050 \pm 0.007$ | 0.69 |
| GCN | $0.066 \pm 0.005$ | $0.040 \pm 0.005$ | – | $0.067 \pm 0.008$ | $0.045 \pm 0.008$ | 0.64 | $0.057 \pm 0.007$ | $0.039 \pm 0.007$ | 0.72 |
| GIN | $0.065 \pm 0.006$ | $0.038 \pm 0.004$ | – | $0.070 \pm 0.007$ | $0.048 \pm 0.008$ | 0.53 | $0.059 \pm 0.007$ | $0.040 \pm 0.007$ | 0.71 |
| EGNN | $0.065 \pm 0.005$ | $0.036 \pm 0.004$ | – | $0.065 \pm 0.007$ | $0.043 \pm 0.009$ | 0.65 | $0.063 \pm 0.006$ | $0.038 \pm 0.006$ | 0.75 |
| MLP-Div | $0.066 \pm 0.005$ | $0.041 \pm 0.005$ | 0.81 | $0.034 \pm 0.006$ | $0.016 \pm 0.005$ | 0.93 | $0.037 \pm 0.005$ | $0.019 \pm 0.004$ | 0.90 |
| GCN-Div | $0.071 \pm 0.006$ | $0.048 \pm 0.005$ | 0.81 | $0.034 \pm 0.006$ | $0.017 \pm 0.005$ | 0.92 | $0.037 \pm 0.007$ | $0.019 \pm 0.004$ | 0.90 |
| Bil-MLP | $0.069 \pm 0.005$ | $0.038 \pm 0.004$ | 0.79 | $0.029 \pm 0.006$ | $0.011 \pm 0.002$ | 0.94 | $0.033 \pm 0.006$ | $0.016 \pm 0.003$ | 0.93 |
| Bil-GCN | $0.062 \pm 0.005$ | $0.034 \pm 0.003$ | 0.82 | $0.029 \pm 0.006$ | $0.011 \pm 0.002$ | 0.94 | $0.032 \pm 0.006$ | $0.016 \pm 0.003$ | 0.93 |
| FlowSymm | $\mathbf{0.057 \pm 0.004}$ | $\mathbf{0.028 \pm 0.002}$ | **0.85** | $\mathbf{0.026 \pm 0.005}$ | $\mathbf{0.010 \pm 0.001}$ | **0.96** | $\mathbf{0.029 \pm 0.005}$ | $\mathbf{0.014 \pm 0.002}$ | **0.95** |

### 4.3 ABLATION STUDY: BASIS SIZE & ROLE OF ATTENTION

We conducted four ablation experiments: (i) the number of basis vectors, (ii) the effect of group-action setup, and (iii) the learning of weights on basis vectors through attention (iv) the effect of bilevel optimization. We report the RMSE on the *held-out* edges of each dataset (Table 2.) for each study. For (i), we sweep $k \in \{64, 128, 256, 512\}$ while keeping all other hyper-parameters fixed. Performance improves steadily up to $k = 256$ and then saturates, indicating that 256 basis vectors already capture the relevant divergence-free variability. Larger bases offer no additional benefit but incur higher memory cost, hence we adopt $k = 256$ as the default. (ii) Comparison of FLOWSYMM with Bil-GAT, the ablated variant without group-action modeling, reveals a clear and consistent performance gap. These consistent improvements show that group-action basis provides a meaningful accuracy boost over a standard attention-only model, confirming that physics-informed symmetry constraints help guide more precise flow reconstructions.(iii) Lastly, we keep the basis size fixed at $k = 256$ but remove the attention mechanism: 256 group actions are sampled uniformly at random and combined with a *single* global weight. Compared with the full FLOWSYMM ($k = 256$), this variant increases RMSE by $+5\%$ on **Traffic** , $+11.5\%$ on **Power**, and $+6.8\%$ on Bike). The gap shows the performance gains that arise from the *edge-wise attention weights.* (iv) Finally, removing the bilevel optimization framework (GAT) results in a substantial degradation in performance.

Table 2: **RMSE results across ablation variants.** Increasing the basis size from 64 to 256 reduces error, after which it saturates. Removing the group-action module (Bil-GAT) or using uniform (non-attention) weights both increase RMSE.

| Variant | Traffic | Power | Bike |
|---|---|---|---|
| FlowSymm ($k = 64$) | 0.060 | 0.029 | 0.031 |
| FlowSymm ($k = 128$) | 0.058 | 0.028 | 0.030 |
| FlowSymm ($k = 256$) | **0.057** | **0.026** | **0.029** |
| FlowSymm ($k = 512$) | 0.057 | 0.026 | 0.029 |
| Bil-GAT (no group-action) | 0.062 | 0.029 | 0.032 |
| GAT (No bilevel / implicit) | 0.067 | 0.065 | 0.062 |
| No-attention ($k = 256$) | 0.060 | 0.029 | 0.031 |

## 5 DISCUSSION

Our experiments demonstrate that FLOWSYMM achieves the strongest performance across three qualitatively different flow domains, improving RMSE by $8$–$10\%$ over the previous state-of-the-art (Silva et al., 2021). The gains are consistent across MAE and correlation, confirming that the symmetric, physics-aware corrections learned by the model not only reduce average error but also better reproduce the true spatial pattern of flows. The ablation study in Sec4.3 shows two key effects. First, enlarging the basis from $k = 64$ to $k = 256$ steadily reduces error, then saturates, indicating that a few hundred group actions already capture the most important degrees of freedom. Second, replacing learned attention with random weights or removing the group module altogether (Bil-GAT) incurs a clear penalty. These trends support our design choice: attention lets the solver place corrective mass where local topology and features call for it, while the group basis guarantees that such corrections preserve the global conservation pattern. Although a systematic qualitative study is beyond the scope of this work, the sparsity implied by that makes it feasible to inspect individual high-scoring group actions and trace how they redistribute flow while remaining divergence-free. Because every action lies in $\ker B$, these adjustments are directly interpretable as redistributions that respect physical laws. Viewed another way, FLOWSYMM performs a meta-search over the divergence-free flow manifold: it first enumerates a truncated basis of admissible group actions and then uses attention to navigate and select optimal corrections. Intuitively, FLOWSYMM first enumerates a basis of admissible adjustments and then uses a GNN to learn a feature-based attention mechanism that selects an optimal combination of these adjustments. This meta-search framing highlights how our model adaptively explores physics-preserving adjustments, rather than relying on a fixed regularizer, to find the best flow completion.

## 5.1 LIMITATIONS

Despite the discussed benefits, there are some limitations. We train on single–shot graphs and do not model temporal dynamics. Extending the approach with recurrent encoders or time–coupled group actions is a promising direction. While $k = 256$ sufficed in our graphs, very large or highly cyclic networks may require more basis vectors, increasing memory; adaptive selection strategies could mitigate this. Additionally, we follow a randomized edge hold-out protocol, and if real sensor placements exhibit systematic biases (e.g., clustered in urban centers), performance could degrade under different masking patterns. Furthermore, in cases where unobserved cycles are fully determined by known injections, our basis construction may overstate the degrees of freedom; however, the attention mechanism mitigates this by learning to suppress redundant basis vectors. Lastly, the inner solver is convex, but the overall bilevel training is non-convex and may converge to suboptimal local minima, making hyperparameter initialization and tuning important.

## 5.2 FUTURE WORK

Beyond addressing the limitations above, we see two immediate extensions: (i) relax the known-injection assumption by *jointly* inferring the net nodal injections $c$ alongside the missing flows, enabling operation in settings with uncertain or partially observed source/sink measurements; and (ii) model *temporal dynamics* by extending the encoder or group-action module with recurrent or graph-temporal architectures, so that spatio-temporal correlations across successive snapshots inform more accurate imputations.

We also emphasize that FlowSymm is **modular and complementary**: any stronger encoder (including conservation-aware message-passing) can replace the GATv2 block within our pipeline, while the group-action layer and implicitly-differentiated refinement continue to provide the physics guarantees. The same symmetry-preserving, basis-driven three-step strategy readily adapts to other missing-data problems governed by linear constraints—for example, inferring heat or concentration fields in diffusion processes (via Laplacian group actions), estimating gaps in electrical potential fields, or imputing state variables in compartmental epidemiological models. Each application requires crafting an appropriate group action and solver but follows the same overall recipe.

In summary, our results suggest that modest, physically motivated modifications to the basis, attention weights, and the implicitly differentiated refinement suffice to push performance beyond more generic GNN or bilevel formulations, while keeping the model transparent and controllable for practitioners in transportation, energy, and mobility planning.

## 5.3 ACKNOWLEDGEMENTS

This material is based upon work supported by the National Science Foundation under grant no. IIS-2229876 and is supported in part by funds provided by the National Science Foundation, by the Department of Homeland Security, and by IBM. Any opinions, findings, and conclusions or recommendations expressed in this material are those of the author(s) and do not necessarily reflect the views of the National Science Foundation or its federal agency and industry partners.

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

## APPENDIX A: NOTATION LIST

| Symbol | Meaning |
|---|---|
| $n,\ m$ | Number of vertices / edges in the graph. |
| $\mathcal{G} = (\mathcal{V}, \mathcal{E}, X)$ | Input graph: vertex set $\mathcal{V}$, edge set $\mathcal{E}$, edge features $X$. |
| $\mathcal{E}_{\text{obs}},\ \mathcal{E}_{\text{miss}}$ | Observed / missing flow subsets. |
| $m_{\text{obs}} = |\mathcal{E}_{\text{obs}}|,\ m_{\text{miss}} = |\mathcal{E}_{\text{miss}}|$ | Number of observed / missing flows. |
| $r = m_{\text{miss}} - n + 1$ | Dimension of full divergence-free group-action space. |
| $B \in \{-1, 0, 1\}^{n \times m}$ | Oriented incidence matrix of $\mathcal{G}$. |
| $B_{\text{obs}} = B\, S_{\text{obs}}^{\top},\ B_{\text{miss}} = B\, S_{\text{miss}}^{\top}$ | Incidence restricted to observed / missing flows. |
| $S_{\text{obs}} \in \{0, 1\}^{m_{\text{obs}} \times m},\ S_{\text{miss}} \in \{0, 1\}^{m_{\text{miss}} \times m}$ | Selector matrices for observed / missing flows. |
| $P_{\text{bal}} = I_m - B^{\top}(BB^{\top})^{\dagger}B$ | Projector onto $\ker B$. |
| $P_{\text{miss}} = \text{diag}(\mathbf{1}_{e \in \mathcal{E}_{\text{miss}}})$ | Mask onto missing-edge coordinates. |
| $\hat{f} \in \mathbb{R}^m$ | Partially observed flow vector (zeros on $\mathcal{E}_{\text{miss}}$). |
| $\hat{f}^{\text{obs}} = S_{\text{obs}}\hat{f}$ | Observed portion of $\hat{f}$. |
| $c \in \mathbb{R}^n$ | Net nodal injection ($c_i < 0$: sources, $c_i > 0$: sinks). |
| $f^{(0)} = \hat{f} + S_{\text{miss}}^{\top}\delta^{(0)}$ | Minimum-norm balanced completion ($Bf^{(0)} = c$). |
| $\delta^{(0)} = S_{\text{miss}}f^{(0)}$ | Missing-component in $f^{(0)}$. |
| $U \in \mathbb{R}^{m \times k}$ | Truncated orthonormal basis for $\mathcal{A}$ ($k \leq r$). |
| $u^{(i)}$ | $i$th column of $U$, a divergence-free basis vector. |
| $k$ | Number of retained basis vectors (default 256). |
| $\mathcal{A} = \{\, u : B\,u = 0,\ u_e = 0\ \forall e \in \mathcal{E}_{\text{obs}} \,\}$ | Admissible divergence-free adjustments. |
| $L$ | Number of stacked GATv2 layers. |
| $X \in \mathbb{R}^{m \times d}$ | Edge-feature matrix (one row per edge). |
| $H = \text{GATv2}_{\theta}(\mathcal{G}, X) \in \mathbb{R}^{m \times d'}$ | Edge embeddings; $H_e \in \mathbb{R}^{d'}$ for edge $e$. |
| $d, d'$ | Input / hidden dimensions of GATv2. |
| $w_i \in \mathbb{R}^{d'}$ | Trainable attention vector for basis $i$. |
| $q_{e,i} = (w_i^{\top} H_e)\,|u_e^{(i)}|$ | Compatibility score between edge $e$ and basis $i$. |
| $Q = [q_{e,i}] \in \mathbb{R}^{m_{\text{miss}} \times k}$ | Matrix of all compatibility scores. |
| $s_i = \frac{1}{m_{\text{miss}}}\sum_{e \in \mathcal{E}_{\text{miss}}} q_{e,i}$ | Aggregated score for basis $i$. |
| $\alpha_{\theta} = \text{softmax}(s) \in \Delta^{k-1}$ | Attention weights over columns of $U$. |
| $\Delta = U\,\alpha_{\theta}$ | Divergence-free group-action. |
| $f_{\theta}^{\text{cand}} = f^{(0)} + \Delta$ | Candidate balanced flow. |
| $\delta_{\theta}^{\text{attn}} = S_{\text{miss}}f_{\theta}^{\text{cand}}$ | Candidate missing values after attention. |
| $\hat{c} = B_{\text{obs}}\hat{f}^{\text{obs}}$ | Empirical imbalance from observations. |
| $\lambda > 0$ | Learnable Tikhonov regularization weight. |
| $\delta_{\theta}^{\text{tik}} = \arg\min_{\delta} \|B_{\text{miss}}\delta + B_{\text{obs}}f_{\theta}^{\text{cand obs}} - \hat{c}\|^2 + \lambda\|\delta - \delta_{\theta}^{\text{attn}}\|^2$ | Tikhonov-refined missing values. |
| $\tilde{f}_{\theta} = f_{\theta}^{\text{cand}} + S_{\text{miss}}^{\top}(\delta_{\theta}^{\text{tik}} - \delta_{\theta}^{\text{attn}})$ | Final model output. |
| $K$ | Number of cross-validation folds. |
| $g^{\text{true},(k)}$ | Ground-truth flows on held-out edges in fold $k$. |
| $S_{\text{val}}^{(k)}$ | Selector matrix for validation edges in fold $k$. |

APPENDIX B: EVALUATION METRIC DEFINITIONS

Let $\mathcal{E}_{\text{val}} \subseteq \mathcal{E}$ be the set of held-out edges of size $m' = |\mathcal{E}_{\text{val}}|$. For each $e \in \mathcal{E}_{\text{val}}$, let $\tilde{f}_e$ and $f_e$ denote the model's predicted and the ground-truth flow, respectively. We report four metrics:

**Root Mean Squared Error (RMSE):**

$$\text{RMSE} \;=\; \sqrt{\frac{1}{m'} \sum_{e \in \mathcal{E}_{\text{val}}} \left(\tilde{f}_e - f_e\right)^2}.$$

**Mean Absolute Error (MAE):**

$$\text{MAE} \;=\; \frac{1}{m'} \sum_{e \in \mathcal{E}_{\text{val}}} \left|\tilde{f}_e - f_e\right|.$$

**Pearson Correlation (Corr):**

$$\text{Corr} \;=\; \frac{\text{cov}(\tilde{f},\, f)}{\sigma(\tilde{f})\,\sigma(f)},$$

where

$$\text{cov}(\tilde{f},\, f) = \frac{1}{m'} \sum_{e \in \mathcal{E}_{\text{val}}} (\tilde{f}_e - \overline{\tilde{f}})(f_e - \overline{f}), \quad \sigma(\tilde{f}) = \sqrt{\frac{1}{m'} \sum_e (\tilde{f}_e - \overline{\tilde{f}})^2},$$

and $\overline{\tilde{f}} = \frac{1}{m'} \sum_e \tilde{f}_e, \overline{f} = \frac{1}{m'} \sum_e f_e$.

APPENDIX C: EXPERIMENTS DETAILS

**Traffic:** We use Los Angeles County's road network as a directed graph in which vertices correspond to junctions and edges to individual road segments. The underlying topology is extracted from OpenStreetMap and then compressed by merging any maximal sequence of non-branching segments into a single edge, yielding $5\,749$ vertices and $7\,498$ directed links. Silva et al. (2021); OpenStreetMap contributors (2017) Traffic flows are given by daily average vehicle counts recorded in 2018 by Caltrans PeMS (Performance Measurement System) California Department of Transportation (2018) sensors placed on major highways; each sensor is matched to its nearest compressed edge using geospatial coordinates and auxiliary sensor metadata. In total, $2,879$ edges (38% of the network) carry flow measurements. Every edge is annotated with an 18-dimensional feature vector that includes: the latitude/longitude of its upstream and downstream endpoints; the number of lanes; the posted speed limit; a hot encoding of highway type (e.g., motorway, motorway link, trunk); and three graph-based statistics such as the in-degree and PageRank centrality of the source vertex, and the out-degree of the target vertex. We note that, due to occasional sensor malfunctions and imperfect edge–sensor matchings, this dataset exhibits noisier measurements than our Power and Bike benchmark, making flow estimation particularly challenging. OpenStreetMap is licensed under the Open Database License (ODbL) 1.0. Caltrans PeMS data are publicly available under Caltrans' data sharing policy (CC BY 4.0).

**Power:** We use the continental European transmission system as an undirected graph whose vertices are electrical buses and whose edges include both physical transmission lines and auxiliary bus connectors for generation and consumption. Active power flows (in MW) are taken from a single snapshot generated by PyPSA-Eur Brown et al. (2018); Hörsch et al. (2018), a Python toolbox that solves optimal linear power-flow problems using historical load and generation data under default settings. The resulting network comprises 2,048 buses and 2,729 edges. Each edge is equipped with a 14-dimensional feature vector: for line edges, we include reactance $x$, resistance $r$, nominal capacity $s_{\text{nom}}$, a binary flag indicating whether $s_{\text{nom}}$ is extendable, the capital cost of capacity expansion, line length, number of parallel circuits, and optimized capacity $s_{\text{nom}}^{\text{opt}}$; for bus edges, we encode the control strategy (PQ, PV, or slack) via one-hot encoding. A binary indicator distinguishes line versus bus edges, and all categorical attributes are represented with one-hot vectors Silva et al. (2021). PyPSA-Eur is released under multiple licenses: original source code under the MIT License, documentation under CC-BY-4.0, configuration files mostly under CC0-1.0, and data files under CC-BY-4.0.

**Bike:** We build a micromobility flow graph from the public *Citi Bike* system data for March 2025 Citi Bike NYC (2025). After discarding trips with missing station IDs and aggregating intra-month usage, we create a vertex for active locations (1 966 total). For every station pair $(i, j)$ we sum the trip counts *in both directions* and retain the $2,647$ most-frequent pairs, resulting in an *undirected* edge set. Symmetrizing in this way removes spurious directionality driven by one-way streets while preserving the capacity-planning signal. Edge flows are the average daily bike transfers over the month. Nine edge features are retained: latitude/longitude of both locations, duration of trip, rideable type (classic / electric), fraction of trips by members, fraction of trips in peak hours (5-7PM and 7-9AM), and distance between locations. The data is made available for public use, though specific licensing terms are not explicitly stated. We have ensured compliance with all usage guidelines provided by Citi Bike.

Table 3: FlowSymm - Standard deviations (over 10 folds) for RMSE and MAE on each dataset.

| Dataset | RMSE | MAE |
|---------|------|------|
| Traffic | 0.004 | 0.002 |
| Power | 0.005 | 0.001 |
| Bike | 0.005 | 0.002 |

**Stability across folds (Table 3)** We report the standard deviation of both RMSE and MAE over our 10 cross-validation folds. Across all three datasets, FLOWSYMM exhibits very low variability,RMSE fluctuates by only 0.004–0.005 and MAE by 0.001–0.002, indicating that its performance is highly consistent regardless of which edges are held out.

## APPENDIX D: SCALABILITY AND IMPLEMENTATION DETAILS

We present a detailed complexity analysis of the components of FlowSymm. The computational cost of our method is driven by the GNN encoder and the two algebraic stages: creating an initial balanced anchor and the final Tikhonov refinement. In our analysis, we simplify the notation by treating fixed-size hyperparameters as constants. Specifically, the hidden dimension of the encoder $D$ and the number of basis vectors $k = 256$ do not grow with the size of the graph. Furthermore, the number of nonzero entries in the incidence matrix, $\mathrm{nnz}(B)$, is simply $2m$ for a graph with $m$ edges, so we consider its complexity to be $\mathcal{O}(m)$.

For numerical stability, our current implementation utilizes dense matrices for the core solver steps, which dictates the overall complexity.

- **Balanced Anchor:** Requires computing a dense pseudoinverse on the $n \times n$ matrix $B_{\mathrm{miss}} B_{\mathrm{miss}}^\top$, which costs $\mathcal{O}(n^3)$ time.

- **Tikhonov Refinement:** Involves forming and solving the $m_{\mathrm{miss}} \times m_{\mathrm{miss}}$ normal equations. Building the matrix $B_{\mathrm{miss}}^\top B_{\mathrm{miss}}$ costs $\mathcal{O}(m_{\mathrm{miss}}^2 n)$, and the Cholesky decomposition costs $\mathcal{O}(m_{\mathrm{miss}}^3)$.

The total per-iteration cost is the sum of the encoder, attention, and solver costs. Since the GNN and attention steps are linear in $m$, the overall complexity is dominated by the dense solvers:

$$\mathcal{O}(m + n^3 + m_{\mathrm{miss}}^3).$$

The complexity of the dense solvers shifts significantly in a scenario where the number of missing edges $m_{\mathrm{miss}}$ is assumed to be a fixed constant that does not grow with the overall size of the network. Under this condition, any term dependent on $m_{\mathrm{miss}}$ becomes constant. Specifically, the cost of the Cholesky decomposition, $\mathcal{O}(m_{\mathrm{miss}}^3)$, becomes a constant factor that can be disregarded in the asymptotic analysis. Similarly, the cost of forming the normal equations matrix, $\mathcal{O}(m_{\mathrm{miss}}^2 n)$, simplifies to $\mathcal{O}(n)$. The total per-iteration complexity, $\mathcal{O}(m + n^3 + m_{\mathrm{miss}}^3)$, therefore reduces to $\mathcal{O}(m + n^3)$. Consequently, the overall complexity is dominated by the balanced anchor step, and the practical bottleneck becomes the number of nodes in the graph, simplifying the complexity to $\mathcal{O}(n^3)$. The dense implementation is not fundamental to our method. By using iterative solvers, the scaling can be dramatically improved.

- Both the anchor and refinement steps can be reformulated as sparse linear systems solvable with the Conjugate Gradient (CG) method. The $\mathcal{O}(n^3)$ cost of the anchor step is a result

of direct matrix inversion. The CG method avoids this cubic bottleneck by re-framing the problem as an iterative solver. Each iteration requires only matrix-vector products, which can be computed in $\mathcal{O}(m)$ time without ever using the dense $n \times n$ matrix, thus reducing the overall complexity of this step to linear.

- These solvers operate via matrix-vector products (e.g., $v \mapsto (B_{\mathrm{miss}}^{\top} B_{\mathrm{miss}} + \lambda I)v$), which can be computed efficiently without materializing dense matrices, costing only $\mathcal{O}(m)$ per iteration.

Assuming the number of CG iterations $T_{\mathrm{CG}}$ is effectively constant for well-conditioned graphs, the complexity of each solver step becomes linear. The total per-iteration cost for this ideal implementation simplifies to:

$$\mathcal{O}(m).$$

This shows that FlowSymm's architecture is compatible with methods that scale linearly with the number of edges, making it suitable for much larger graphs with appropriate engineering.

# A    APPENDIX E: ADDITIONAL EXPERIMENTAL ANALYSES

In this section, we provide detailed results from three additional experiments conducted to verify the robustness, physical consistency, and architectural superiority of FLOWSYMM.

## A.1    SENSITIVITY TO NOISE IN NODAL INJECTIONS ($c$)

A key input to our method is the net nodal injection vector $c$, which is used to compute the initial balanced anchor $f^{(0)}$. To evaluate the robustness of FLOWSYMM to measurement errors or uncertainty in $c$, we conducted a sensitivity analysis. We trained the model using the ground-truth $c$, and at inference time, we injected Gaussian noise into $c$ (scaled by percentages of the standard deviation of $c$).

Table 4 reports the $L_2$-norm of the anchor ($f^{(0)}$), the $L_2$-norm of the learned group-action correction ($\Delta$), and the final Test RMSE. The results show that while noise in $c$ significantly perturbs the anchor, the model's learned correction $\Delta$ (which relies on edge features) remains stable. Consequently, the final prediction performance is exceptionally robust, with negligible degradation in RMSE even at $50\%$ noise levels.

Table 4: Sensitivity analysis of FLOWSYMM under varying levels of noise added to the injection vector $c$ at test time. Despite significant drift in the anchor norm, the final RMSE remains stable.

| Dataset | Noise Level | Anchor Norm ($\|f^{(0)}\|_2$) | Action Norm ($\|\Delta\|_2$) | Test RMSE |
|---|---|---|---|---|
| **Traffic** | 0% | 17.45 | 5.33 | 0.057 |
| | 15% | 17.54 | 5.33 | 0.057 |
| | 35% | 18.21 | 5.33 | 0.058 |
| | 50% | 18.45 | 5.33 | 0.060 |
| **Power** | 0% | 2.19 | 1.38 | 0.026 |
| | 15% | 2.19 | 1.38 | 0.026 |
| | 35% | 2.20 | 1.38 | 0.027 |
| | 50% | 2.21 | 1.38 | 0.028 |
| **Bike** | 0% | 2.27 | 1.05 | 0.029 |
| | 15% | 2.27 | 1.05 | 0.029 |
| | 35% | 2.26 | 1.05 | 0.029 |
| | 50% | 2.32 | 1.05 | 0.030 |

## A.2    COMPARISON WITH "PREDICT-THEN-PROJECT" BASELINE

To validate the benefit of learning *within* the valid group-action subspace versus projecting *onto* it, we implemented a "Predict-then-Project" (PnP) baseline. This baseline trains an identical GATv2 encoder to predict an unconstrained correction $z \in \mathbb{R}^m$, which is then projected onto the feasible subspace via $\Delta = P_{\mathcal{A}}z$.

As shown in Table 5, the PnP baseline performs significantly worse than FLOWSYMM and, notably, worse than the physics-agnostic MLP. This confirms that learning an unconstrained vector and

hoping for a successful projection is a difficult learning task with poor gradient signals, whereas FLOWSYMM's approach of learning coefficients for a valid basis is far more effective.

Table 5: RMSE comparison between FLOWSYMM, a physics-agnostic MLP, and the Predict-then-Project (PnP) baseline. Lower is better.

| Model | Traffic | Power | Bike |
|---|---|---|---|
| MLP (No Physics) | 0.083 | 0.083 | 0.073 |
| PnP Baseline (Projected) | 0.201 | 0.127 | 0.145 |
| **FlowSymm (Ours)** | **0.057** | **0.026** | **0.029** |

### A.3 ANALYSIS OF PHYSICAL CONSISTENCY (DIVERGENCE RESIDUALS)

Finally, we evaluate the physical adherence of the models by measuring the $L_2$-norm of the divergence residual, $||B\tilde{f} - c||_2$. Table 6 compares FLOWSYMM against both physics-agnostic (MLP, GCN) and physics-aware (Min-Div, Bil-GCN) baselines.
The results demonstrate that FLOWSYMM achieves the lowest (or tied for lowest) divergence residual across all datasets. Unlike the MLP and GCN, which violate conservation laws significantly (high residuals), FLOWSYMM maintains strict physical consistency while simultaneously achieving state-of-the-art reconstruction accuracy.

Table 6: Final Divergence Residual ($||B\tilde{f} - c||_2$) on the test set. Lower values indicate better adherence to physical conservation laws.

| Model | Traffic | Power | Bike |
|---|---|---|---|
| MLP | 5.69 | 2.73 | 2.75 |
| GCN | 5.71 | 2.76 | 2.77 |
| Min-Div | 2.94 | 2.45 | 2.40 |
| Bil-GCN | 2.83 | 2.43 | 2.38 |
| **FlowSymm (Ours)** | **2.81** | **2.42** | **2.37** |

## APPENDIX F: REPRODUCIBILITY STATEMENT

To ensure our results are fully reproducible, we provide a comprehensive set of resources. The complete source code for FLOWSYMM, including data preprocessing scripts, model implementations, and evaluation notebooks, is included in the supplementary material and will be made publicly available upon publication. We train all models with the Adam optimizer for at most 10 epochs, using a base learning rate (`outer_lr`) of $10^{-2}$. We selected a 10 epochs for our end-to-end bilevel training because in preliminary experiments the validation loss consistently plateaued by epoch 8 across all folds, and extending beyond 10 yielded negligible gains at substantially higher compute cost. Early stopping is applied with patience 10 on the held-out loss. Our GATv2 encoder uses an 18/14/9-dimensional input (depending on dataset), a hidden size of `n_hidden=16`, `num_heads=4` attention heads. All results use 10-fold cross-validation (`n_folds=10`). All experiments were run on a dual-socket Intel Xeon CPU Max 9470 server and a single partition of NVIDIA H200 (18G). For the bilevel-MLP and bilevel-GCN baselines, we directly adopted the hyperparameters provided in Silva et al. Silva et al. (2021), without additional tuning, to match the original experimental setup. We tuned all other models by minimizing RMSE via a grid search over learning rates $\{10^0, 10^{-1}, 10^{-2}, 10^{-3}\}$ and hidden-layer widths $\{4, 8, 16\}$. For the Min–Div baseline we ran up to 3,000 iterations, and for MLP and GCN up to 5,000 iterations, employing early stopping if no improvement was seen for 10 consecutive steps. In order to reproduce the *flow-completion* experiments, one can begin by opening the notebook `flowpred_pipeline.ipynb`. This file is a streamlined version of the raw command-line pipeline: it stitches together every preprocessing step, model-initialization block, and training loop in exactly the order described in this report. Once the required Python packages have been installed with `pip install`), no further configuration is necessary. Note that the code embedded in `flowpred_pipeline.ipynb` is configured to demonstrate *one* cross-validation

fold so that a first pass finishes quickly; one can switch to a full 10-fold evaluation using ten fold function from the evaluation.

