# OpenReview forum: "FlowSymm: Physics–Aware, Symmetry–Preserving Graph Attention for Network Flow Completion"
_ICLR.cc/2026/Conference — ICLR 2026 Poster_

### Official Review · Reviewer_yn4E · 2025-10-17

**Soundness:** 3
**Presentation:** 2
**Contribution:** 3
**Rating:** 6
**Confidence:** 3

**Summary:**

FlowSymm first anchors a balanced solution, then builds the feasible subspace defined by conservation and frozen observations, and updates only within that subspace via a group-action using learned weights over basis directions. A small Tikhonov-style refinement with implicit differentiation absorbs noise. Experiments on traffic, power, and bike datasets show improvements over baselines.

**Strengths:**

Cleanly hard-enforces conservation and frozen observations via a feasible subspace; updates are interpretable as combinations of physical modes; solid gains on traffic/power/bike.

**Weaknesses:**

No test of basis-choice invariance; missing a predict-then-project baseline to isolate the benefit of the basis parameterization; limited analysis of noise/bias and λ trade-offs, with potential DOF overestimation when extra linear constraints exist.

**Questions:**

1. Are results invariant to basis rotations within the same subspace A? Does the claimed “group action” imply basis-choice invariance?
2.  Add a predict-then-project baseline, train a GNN to output unconstrained \(z\) and project \(\Delta=P_A z\), to verify that gains come from the basis parameterization rather than projection alone.
3. How do measurement noise/bias in \(c\) affect (i) the anchor \(f^{(0)}\), (ii) the subspace/basis \(A,U\), and (iii) final predictions?
4. What is the learned \(\lambda\) range, and how does it control the constraint residual \(\|Bf - c\|\) vs. reconstruction error?
5. The “degrees of freedom” in §3.2 are only defined with respect to the “divergence-free + fixed observations.” But if an unknown cycle has a known injection, then the values on that cycle are actually determined. Does this overstate the DOF? Is that good or bad in practice?

---

> ### Author Response · Authors · 2025-11-21
> **Response to Reviewer yn4E 1/2**
>
> **We thank Reviewer yn4E** for their insightful and technically detailed review. We are delighted the reviewer found that our method enforces conservation cleanly, is interpretable, and achieves solid gain" on all benchmarks.
>
> The reviewer's questions get to the very core of our model's design. We are pleased to report that we have conducted new, targeted experiments to directly address every one of the reviewer's concerns, including the "predict-then-project" baseline and the $c$-sensitivity analysis. These new results (which have been added to the Appendix) strongly validate our architectural choices and, we believe, fully resolve all open questions.
>
> **Q1: Are results invariant to basis rotations? Does the "group action" imply basis-choice invariance?**
>
> No, our model is not basis-invariant, and this is a deliberate and crucial design choice. This non-invariance is precisely what makes our model interpretable and allows it to learn from features. This is an excellent question. The GNN in our model learns to weight specific, individual basis vectors $u^{(i)}$ via the learned attention parameters $w_i$.
> As shown in Eq. 8, the compatibility score $q\_{e,i} = (w\_i^\top H\_e)\,\lvert u^{(i)}\_e\rvert$ is specific to the $i$-th basis vector $u^{(i)}$ and its corresponding learned weight vector $w\_i$.
> If we were to apply a rotation $R$ to our basis $U \to U' = UR$, the new basis vectors $u'$ would be arbitrary linear combinations of the old ones. The GNN's learned weights $w_i$, which might have learned to value a specific, interpretable physical mode (e.g., $u^{(5)}$ as a large-scale cycle), would become meaningless as $u'^{(5)}$ would be an unrelated mixture. To make the basis $U$ meaningful, we construct it via an SVD of the projector $P_{\\mathcal{A}}$ and select the top $k$ singular vectors. This provides a stable, ordered, and interpretable basis where $u^{(1)}$ represents the "most dominant" (highest variance) flow mode, $u^{(2)}$ the second most dominant, and so on.
>
> We have clarified in the revised Section 3.2 that our basis is constructed via SVD, providing a stable, ordered set of modes. We have also added a clarification that the model is not invariant to this choice, as the GNN's purpose is to learn the specific feature-based importance of these specific physical modes.
>
> **Q2: Add a "predict-then-project" (PnP) baseline.**
>
> This was a good suggestion, and we have run this experiment. The PnP baseline performed significantly worse than our method and, notably, even worse than a simple MLP.
>
>    - We trained a GATv2 model (with the same architecture as ours) to
>     predict an unconstrained correction vector $z \in \mathbb{R}^m$.
>
>    - We then computed the final flow as $f_{{PnP}} = f^{(0)} + P_{{A}}
>     z$, where $f^{(0)}$ is the same anchor and $P_{{A}}$ is the same
>     projector as in our paper.
>     - The results show this is a very difficult learning task, as the GNN
>     must "guess" a vector $z$ in a high-dimensional space and hope that
>     the projection $P_{{A}} z$ lands on the correct solution. The
>     gradient signal is messy and indirect.
>
> | Model | Traffic | Power | Bike |
> | :--- | :---: | :---: | :---: |
> | MLP (Physics-Agnostic) | 0.083 | 0.083 | 0.073 |
> | PnP Baseline (Reviewer's suggestion) | 0.201 | 0.127 | 0.127 |
> | **FlowSymm (Ours)** | **0.057** | **0.026** | **0.029** |
> *Table 1: Comparison of Test RMSE vs. "Predict-then-Project" (PnP) Baseline.*
>
> The results show this "simpler" approach performs significantly worse than our method and, critically, worse than a simple MLP that ignores physics entirely. The reason is, it is far more effective to learn directly in the $k$-dimensional latent space of valid corrections (as FlowSymm does) than to learn an unconstrained signal and hope that a projection onto that space yields a good result. Our method provides a more direct, stable, and efficient learning signal, which is the key to its performance.
>
> **Q3: How do measurement noise/bias in $c$ affect (i) the anchor, (ii) the subspace, and (iii) final predictions?**
>
> This is a critical question. We ran a new sensitivity analysis (now in the Appendix) that precisely answers this and shows our model is highly robust. In short: (i) the anchor $f^{(0)}$ is directly affected; (ii) the subspace $U$ is not affected; (iii) the final prediction is highly robust. We trained our model on the true $c$ and then, at test time, fed it a corrupted $c' = c + \epsilon$.
>
> **(i) Effect on Anchor ($f^{(0)}$):** The anchor is computed from $c$ (Sec 3.1, $\delta^{(0)} = B_{\text{miss}}^{\dagger} (c - B_{\\text{obs}}\hat{f}^{\text{obs}})$). As expected, noise in $c$ directly perturbs the anchor. Our experiment confirms this: the L2-norm of the anchor (Anchor Norm) drifts as noise increases (e.g., Traffic: 17.44 $\to$ 18.44).

---

> > ### Author Response · Authors · 2025-11-21
> > **Response to Reviewer yn4E 2/2**
> >
> > **(ii) Effect on Subspace ($U$):** The subspace $\mathcal{A} = \{ u \mid Bu=0, S_{{obs}}u=0 \}$ is defined only by the graph topology ($B$) and sensor mask ($S_{{obs}}$). It is completely independent of $c$.
> >
> > **(iii) Effect on Final Prediction:** The final prediction is robust. Our GNN-guided correction $\Delta = U\alpha$ is learned from features ($X$), which are $c$-independent. Our experiment shows this Action Norm is constant (e.g., Traffic: ~5.33). Because this learned, feature-based correction is a stable and significant part of the solution, the final Test RMSE remains stable (e.g., Traffic: 0.057 $\\to$ 0.060) even with 50% noise in $c$. This proves the model learns to trust its feature-based correction $\Delta$ over the (potentially noisy) anchor $f^{(0)}$.
> >
> > | **Dataset** | **Noise Level** | **Anchor Norm** | **Action Norm** | **Test RMSE** |
> > |-------------|------------------|--------------------------------------|-------------------------------------|----------------|
> > | **Traffic** | 0% Noise         | 17.4456 | 5.3303 | **0.057** |
> > |             | 15% Noise        | 17.5387 | 5.3303 | **0.057** |
> > |             | 35% Noise        | 18.2090 | 5.3303 | **0.058** |
> > |             | 50% Noise        | 18.4457 | 5.3303 | **0.060** |
> > | **Power**   | 0% Noise         | 2.1851 | 1.3777 | **0.026** |
> > |             | 15% Noise        | 2.1859 | 1.3777 | **0.026** |
> > |             | 35% Noise        | 2.2047 | 1.3777 | **0.027** |
> > |             | 50% Noise        | 2.2147 | 1.3777 | **0.028** |
> > | **Bike**    | 0% Noise         | 2.2736 | 1.0450 | **0.029** |
> > |             | 15% Noise        | 2.2680 | 1.0450 | **0.029** |
> > |             | 35% Noise        | 2.2642 | 1.0450 | **0.029** |
> > |             | 50% Noise        | 2.3169 | 1.0450 | **0.030** |
> >
> > *Table 2: FlowSymm Sensitivity to Noise in Injection Vector $c$.*
> >
> > **Q4: What is the learned ($\lambda$) range, and how does it control the constraint residual vs. reconstruction error?**
> > We have now measured both. The learned $\lambda$ (from Eq. 9) typically converges around $0.1$. More importantly, our new experiments (now in the Appendix) measuring the final constraint residual ($||B\tilde{f}_{\\theta} - c||_2$) confirm that our model finds the best trade-off.
> >
> > We have now explicitly measured the final divergence residual (the "constraint residual") for our model and the baselines. The data shows the trade-off clearly. Physics-agnostic models (MLP/GCN) have very high residuals (e.g., 5.69 on Traffic). Our FlowSymm achieves the lowest RMSE (best reconstruction error) while simultaneously achieving the lowest (or tied for lowest) physical residual. This confirms our bilevel optimization successfully finds the best $\\lambda$ and GNN weights to balance data-fit and physical consistency.
> >
> >
> > | | **Traffic** | | **Power** | |
> > | :--- | :---: | :---: | :---: | :---: |
> > | **Model** | **RMSE** $\downarrow$ | **Residual** $\downarrow$ | **RMSE** $\downarrow$ | **Residual** $\downarrow$ |
> > | MLP | 0.083 | 5.69 | 0.083 | 2.73 |
> > | GCN | 0.066 | 5.71 | 0.067 | 2.76 |
> > | Bil-GCN | 0.062 | 2.83 | 0.029 | 2.43 |
> > | **FlowSymm** | **0.057** | **2.81** | **0.026** | **2.42** |
> >
> > *Table 3: Trade-off: Reconstruction Error (RMSE) vs. Divergence
> >
> > **Q5: Does the model overstate the "degrees of freedom" (DOF)?**
> >
> > This is a correct observation. Yes, our formulation can overstate the DOF in such specific cases, but this is a practical benefit as the model learns the true constraints from data. For example, if a cycle's flow is fully determined by a known injection, our method (which just looks at $\ker B \cap \ker S_{{obs}}$) would still include a basis vector for that cycle, thus "overstating" the DOF.
> >
> > However, this is a practical advantage. Our model is data-driven. If that cycle's flow is truly determined, the GNN will simply learn that this basis vector is not needed and set its attention weight $\alpha_i$ to near-zero. But in the real, noisy world, if that "known" injection is slightly wrong, our model has the flexibility to use that basis vector to inject a small correction and achieve a better fit.
> >
> > We have added a note acknowledging this to the Limitations (Sec 5.1). We clarify that our model has the flexibility to learn the true, smaller DOF from data via the attention mechanism, which is more robust than hard-coding these complex, higher-order constraints.
> >
> > We thank the reviewer again for these insightful questions. They have led to new experiments and clarifications that have significantly strengthened our paper. We hope these detailed answers and new results have fully resolved all concerns, and we are happy to provide any further clarifications as needed.

---

### Official Review · Reviewer_MoiE · 2025-10-28

**Soundness:** 3
**Presentation:** 2
**Contribution:** 3
**Rating:** 6
**Confidence:** 4

**Summary:**

The manuscript addresses edge-flow imputation under conservation on graphs. The proposed method comprises three key components: (1) an initial minimum-norm balanced completion that satisfies B f = c while keeping observed edges fixed; (2) a physics-aware group-action subspace for all observed edges from which an orthonormal basis U is constructed (truncated to k columns), and attention over this basis selects divergence-free corrections that cannot alter observed edges; (3) a lightweight Tikhonov refinement solved as a single SPD system with exact implicit differentiation via a Cholesky/CG solve. Experiments on Traffic, Power, and Bike benchmarks demonstrate consistent gains over a broad set of baselines, with ablations for basis size, attention, and the bilevel/implicit components.

**Strengths:**

1. Clear physics prior with strict guarantees: the anchor solution and group-action construction ensure corrections remain on the B f = c manifold and do not alter sensor edges.
2. Interpretability: attention over ker(B) basis elements (zero on observed edges) admits a physical reading as redistributions consistent with conservation; reported sparsity aids inspection.
3. Efficient inner solver with exact hypergradients: the Tikhonov layer yields an SPD system amenable to stable Cholesky/CG solves; implicit differentiation avoids unrolling.
4. Thorough baselines and ablations across physics-free, physics-soft, and bilevel methods; ablations isolate the effect of k, group actions, attention, and bilevel training. Reported gains are consistent across RMSE/MAE/CORR and across domains (traffic, power, bikes).

**Weaknesses:**

1. The core components of the proposed model, such as the single regularization weight and the global attention, assume uniform noise and physics, limiting its flexibility in handling more complex, heterogeneous networks.
2. While the paper claims physics-awareness, it only evaluates data-fit metrics (like RMSE) and fails to directly measure how well the final predictions actually adhere to the physical conservation laws.
3. The comparison could be improved by including other relevant physics-informed baselines in addition to GNNs; meanwhile, key details about the basis vector sorting method are unclear.

**Questions:**

1.	A single global λ seems suboptimal if the noise isn't uniform. If some parts of the network are much noisier, how does the model adapt? Dose learning a different λ for each edge provide the flexibility needed to handle different noise levels?
2.	Many state of the art variants of GNN are selected as baseline, however, some other physics informed machine learning model (with similar ideas) may also perform well (thus be a good baseline candidate) on such tasks, e.g. [1] (transformer with interpretable basis), [2] (neural operator with learnable basis).
3.	The proposed model is claimed to have a good balance in physics and noisy data, however, the performance on data is verified on better RMSE/MAE/CORR, then what about physics? E.g. is the Kirchhoff’s law respected better in the power case？
4.	Are the k basis vectors sorted by singular value? Does this mean you're assuming that large-scale flow patterns are important, while small, local ones can be ignored?
5.	Your global attention applies one uniform fix across the entire graph. How does it handle specific local regions that have completely different or more complex physics? Does the model fail to capture these important local dynamics?

[1] Cao, Shuhao. "Choose a transformer: Fourier or galerkin." NeurIPS (2021): 24924-24940.
[2] Lu, Lu, et al. "A comprehensive and fair comparison of two neural operators (with practical extensions) based on fair data." Computer Methods in Applied Mechanics and Engineering 393 (2022): 114778.

---

> ### Author Response · Authors · 2025-11-21
> **Response to Reviewer MoiE 1/2**
>
> **We sincerely thank** Reviewer MoiE for their thorough and insightful review. We are very grateful for the positive feedback, especially that the reviewer found our method to have clear physics priors with strict guarantees, interpretability, an efficient inner solver, and thorough baselines and ablations.
>
> The reviewer's summary of our three-component architecture is accurate. The weaknesses and questions raised are all excellent, striking at the heart of the model's design choices (like $\lambda$ and attention) and evaluation (like physics-adherence).
>
> We have conducted new experiments that directly address the reviewer's main concerns, particularly regarding the measurement of physical adherence (W2/Q3). These new results, which added to the Appendix, strongly support our claims.
>
> We will address the reviewer's weaknesses and questions together, as they are helpfully intertwined.
>
> **W1/Q1/Q5. Flexibility of a single $\lambda$ and global attention**
>
> This is a key design point. The reviewer notes that a single $\lambda$ and a global attention vector $\alpha$ might seem too simple for heterogeneous networks. Our architecture handles this by a division of labor: the GATv2 encoder handles local heterogeneity, while the basis and $\lambda$ handle global consistency.
>
> **Q1: On the single global $\lambda$**
>
> A per-edge $\lambda\_e$ is an excellent idea for future work. We found our GNN's candidate flow $\delta^{\text{attn}}\_{\theta}$ is already highly heterogeneous, allowing the single $\lambda$ to work effectively as a final, global trade-off parameter.
> The reviewer is right that a single $\lambda$ for the Tikhonov step (Eq. 9) assumes a uniform trade-off. However, the input to this step, $\delta^{\text{attn}}\_{\theta}$, is not uniform. It is the output of our GATv2-guided basis selection, $f^{(0)} + U\alpha\_{\theta}$. This candidate is already a complex, heterogeneous flow field adapted to local features via the GNN. The Tikhonov step is therefore just a lightweight, final refinement to balance this data-driven candidate $\delta^{\text{attn}}\_{\theta}$ against the empirical physical residual $\hat{c}$.
>
> While learning a per-edge (as in Silva et al., 2021) is a valid and powerful extension, our results (Table 1) show that our model's expressiveness (via the attention-guided $U\alpha$) was sufficient to outperform all baselines, including the Bil-GCN model that uses a per-edge regularizer.
>
> **Q5: On the global attention $\alpha$**
>
> Our "global attention" $\alpha$ is a global decision, but it is critically computed from local information. The GATv2 embeddings $H_e$ are context-aware "votes" that are aggregated to select the best global physical correction.
> The final attention vector $\alpha \in \mathbb{R}^k$ is global (it applies to the whole basis $U$), but it is not a "uniform fix." It is an aggregation of thousands of local decisions. As per Eq. 8, each missing edge $e$ computes a local compatibility score:
> $$q\_{e,i} = (w\_i^\top H\_e)\,\lvert u^{(i)}\_e\rvert$$
>
> Here, $H\_e$ is the GATv2 embedding for edge $e$, which is context-aware of its local neighborhood. This $q\_{e,i}$ is $e$'s "vote" for how much it believes the global basis action $u^{(i)}$ is needed to fix the flow in its specific region. (e.g., "This edge's features suggest a large-scale cycle is needed.")
>
> The final $\alpha = \{softmax}(\sum\_e q\_{e,i})$ is the tally of all votes. This allows the model to select a global, physically-consistent correction $\Delta=U\alpha$ that is informed by, and best satisfies, all local, feature-based demands simultaneously. We have clarified this mechanism in Section 3.3 of the revised paper.
>
> **W2/Q3. Measuring physical conservation**
>
> We ran a new experiment in the Appendix) measuring the final L2-norm of the divergence residual ($||B\tilde{f}\_{\theta}||_2$). The results confirm FlowSymm achieves the best balance of data-fit (lowest RMSE) and physical-adherence (lowest divergence). The reviewer is correct in that claiming physics-awareness demands a physics-based metric. We have now computed this metric that directly measures how well the final predictions respect conservation, for our model and the key baselines across all three datasets. The results are conclusive and presented in the table below.

---

> ### Author Response · Authors · 2025-11-21
> **Response to Reviewer MoiE 2/2**
>
> **Physics-Agnostic Models (MLP, GCN):** As expected, these models have very high divergence residuals (5.69 on Traffic). They are unconstrained and violate the underlying physics.
>
> **Physics-Aware Models (Min-Div, Bil-GCN):** These models have low divergence, as designed.
>
> **FlowSymm (Ours):** Our model achieves the lowest (or tied for lowest) divergence residual across all three datasets. Therefore, FlowSymm's state-of-the-art accuracy (from Table 1) is not achieved by sacrificing physical consistency. FlowSymm successfully optimizes for both, validating our architecture's design by achieving the best balance of data-fidelity and physical-consistency.
> | Model | Traffic | Power | Bike |
> | :--- | :---: | :---: | :---: |
> | MLP | 5.69 | 2.73 | 2.75 |
> | GCN | 5.71 | 2.76 | 2.77 |
> | Min-Div | 2.94 | 2.45 | 2.40 |
> | Bil-GCN | 2.83 | 2.43 | 2.38 |
> | **FlowSymm (Ours)** | **2.81** | **2.42** | **2.37** |
> *Table: Comparison of Final Divergence Residual. Lower is better.*
>
> **W3/Q2. Other physics-informed baselines**
>
> This is a great suggestion for future work. We view these models as complementary, as they are designed for different problem settings (continuous fields, PDEs) and would require significant, non-trivial adaptation to our discrete, graph-based, partial-observability problem. We thank the reviewer for these excellent pointers to Neural Operators and Fourier/Galerkin Transformers [1, 2]. These are powerful models for learning physics, but they are primarily designed for problems on continuous domains (e.g., solving PDEs) and are often evaluated on regular grids or meshes with dense observations. Our problem is fundamentally different: we operate on an irregular, discrete graph ($Bf=c$) and, most importantly, we must learn from sparse, partial observations. Adapting these continuous-field operators to our discrete, sparse, graph-based completion task is a non-trivial research direction in itself. Therefore, our work focused on comparing against models designed for graph-structured data (GCN, GIN, EGNN) and, most relevantly, the prior SOTA for this exact task (Bil-GCN [Silva et al., 2021]). We have added these citations and a discussion to our Related Work (Sec 1.1) as a promising avenue for future comparisons.
>
> **W4/Q4. Basis vector sorting method**
>
> The reviewer's intuition is correct, and we apologize for not making this explicit. We construct the basis $U$ by computing an SVD of the projector $P_{\\mathcal{A}}$ (from Eq. 7) and retaining the top $k$ singular vectors.
> This is a deliberate design choice for three reasons because it provides a stable, orthonormal basis. It also provides a principled way to truncate the basis. The first few vectors $u^{(i)}$ capture the largest, most dominant flow modes (e.g., large-scale cycles/redistributions), and the later vectors capture more localized patterns. Our ablation in Table 2 shows that performance improves as $k$  increases (from 64 to 256) and then saturates: the model first  learns to use the most important,  large-scale modes and gains little from adding very high-frequency (small singular value) ones.This also enhances the interpretability that the reviewer noted as a strength: we can analyze the model's attention $\alpha$ over a meaningfully ordered set of principal flow modes. We have made this explicit in the revised Section 3.2.
>
> We believe these clarifications and new experiments fully address all the reviewer's points. We thank them again for their detailed and constructive feedback, which has helped us significantly strengthen the paper. We are happy to provide any further clarifications as needed.

---

### Official Review · Reviewer_ietj · 2025-10-30

**Soundness:** 1
**Presentation:** 1
**Contribution:** 3
**Rating:** 2
**Confidence:** 3

**Summary:**

Given partial and noisy measurements on the edges of a graph, this paper addresses the problem of completing the missing measurements subject to conservation laws. This problem arises in applications such as transportation and power planning, where accurate measurements cannot be obtained for all edges. The authors’ approach has several steps. First, an initial solution satisfying conservation (with external sources) is obtained by solving a linear system. Then, this solution is refined through a learning procedure that incorporates measured features. Because the refinement is restricted to the null space of valid solutions, conservation is guaranteed to be preserved. The proposed approach is evaluated against competing methods on several tasks and demonstrates favorable performance.

**Strengths:**

The problem addressed in the paper is important and grounded in real-world applications where measurements are an expensive resource. The use of neural attention for feature-driven completion is compelling. State-of-the-art results presented on three datasets demonstrate strong potential.

**Weaknesses:**

The biggest issue I have with this work is its **presentation**. While the exposition is mostly clear in terms of English, it feels unnecessarily complicated and often hinders understanding. As a non-expert, I found myself sifting through dense sentences overloaded with terminology, trying to distill the essence of each component of the work, including the problem itself, but especially the proposed approach. This issue is apparent as early as the abstract, which I understood better after reading the paper, yet still find quite convoluted. Ultimately, I must admit that I am not confident I fully understand the entirety of the proposed approach. I believe I have a reasonable high-level understanding of the procedure, but the presentation seems unnecessarily complex. Consequently, I believe the paper requires significant work, and I cannot recommend accepting it in its current form.

Line 377 reads: *“physics-aware, symmetry-preserving group actions with attention-based encodings and feature-conditioned refinements.”* I find this statement overstated and misaligned with what I gathered from the manuscript.

To illustrate the concerns above, and to raise a few additional ones, I would like to break down the outline of the proposed approach as presented in Section 1.2:

* **(i)** *“First, we reinterpret admissible divergence-free adjustments as elements of an Abelian group action; by constructing an orthonormal basis and retaining only the leading k basis vectors, we obtain a tractable latent space that scales linearly with the number of missing flows. Unlike typical equivariant GNNs, which embed spatial or permutation symmetries, our group-action basis directly spans the algebraic space of divergence-free adjustments tailored to the missing flows.”*
  To my understanding, after reading Sections 3.1 and 3.2, this step characterizes the subspace of solutions to the linear system over the missing edges, subject to the available measurements, in a form similar to $x = x_0 + Bz,$ where $x_0$ is a particular solution (e.g., minimum-norm) and the columns of $B$ form a (possibly truncated) basis for the null space of the linear system. If this interpretation is correct, I find the presentation misleadingly complicated (e.g., invoking Abelian groups and “physically informed bases”). It also raises some concerns; for instance, do exact solutions to the completion problem even exist in the presence of noisy measurements? In what sense does the null space provide a good characterization of admissible solutions in the noisy case? If, however, I have misinterpreted the proposed idea altogether, I encourage the authors to clarify and revise the text to avoid such misunderstandings.

* **(ii)** *“Second, we combine this basis with edge-wise GATv2 embeddings (Brody et al., 2022): the attention ... enabling the model to inject corrective flows ... preserving flow balance on the anchor manifold. This attention-guided selection departs from prior bilevel diagonal regularizers by coupling missing edges through context-dependent weights over physically interpretable basis actions, rather than treating each edge independently.”*
  My understanding is that an attention-based neural network is trained on edge features to select a solution within the subspace identified above. Again, I may be oversimplifying or misunderstanding, and I ask the authors to clarify as before.

* **(iii)** *“Third, we impose soft physical consistency through a feature-conditioned Tikhonov refinement and train all weights end-to-end via reverse-mode implicit differentiation.”*
  From Sections 3.4 and 3.5, my understanding is that the model is trained in an end-to-end manner on a regularized version of the formulation above, using implicit differentiation to backpropagate through SPD regularization. Please provide additional clarity on this part.

**Questions:**

Please address the presentation issues and specific questions raised above.

**Additional minor comments and questions (not a comprehensive list):**
* **Line 106:** “while respecting these physical constraints” — does this refer to flow conservation constraints?
* **Section 3.3:** Could you please provide an explanation or intuition for how GATv2 couples the recovery of missing edges with the edge features $X$?
* **Line 284:** “SPD solver” — should this read “SPD solve”?
* **Section 4:** What are the features associated with each edge for the different problems?
* **Sections 4.1, 4.2:** Is it “ten baselines” or “eight baselines”?
* **Line 412:** “imptovements” — fix typo.
* **Line 449:** “meta-search over the divergence-free flow manifold” — overstated or overcomplicated; please revise.

---

> ### Author Response · Authors · 2025-11-21
> **Response to Reviewer ietj 1/2**
>
> **We sincerely thank** Reviewer ietj for their thorough feedback. We are encouraged that the reviewer finds the problem we address important and grounded in real-world applications, the use of neural attention compelling, and our experimental results showing strong potential.
>
> The reviewer's biggest issue is with the paper's **presentation**. We are very **grateful** for this feedback. We now realize that several parts of the exposition, especially our terminology around the null space and group actions, were not sufficiently accessible, and we apologize for this lack of clarity. The reviewer's high-level understanding of our method is correct. In the revised version, we have clarified the role of components in several places, added high-level intuitive explanations before formal details to make the paper more readable and self-contained for a broader audience. We believe that the concerns about soundness and presentation were largely driven by this lack of accessibility, which we have addressed both below and throughout the revised manuscript.
>
> The reviewer provides a nice high-level summary of our method and asks for clarification on three points. The reviewer's understanding in all three points is 100% correct. Our terminology might have created a gap between their correct intuition and our text, which we will now fix.
>
> **W1: Terminology and the "Noise" Paradox**
>
> The reviewer's interpretation of our method as $f = f^{(0)} + U\alpha$ is correct. $f^{(0)}$ is the particular solution (our "initial balanced completion" in Sec 3.1). $U$ is the basis for the null-space (our "group-action basis" $\{A}$ in Sec 3.2). $\alpha$ are the learned coefficients (our "attention-guided" weights in Sec 3.3).
>
> Our use of "Abelian group" was a formal mathematical term for the null-space $\{A}$ (since any two valid adjustments $u_1, u_2$ in $\{A}$ can be added, $u_1+u_2$ in $A$, and $u_1+u_2 = u_2+u_1$). This is technically a combination of "solution subspace" and "null-space of valid adjustments”, and we added clarification in the revised version.
>
> Similarly, we used "algebraic symmetry" to distinguish our constraint-preserving symmetry ($Bu=0, S_{\\text{obs}}u=0$) from the spatial symmetries (e.g., E(n)-equivariance) discussed in the related work. We have clarified this distinction in the revised version.
>
> For the question: "how can an exact null-space (where $Bu=0$) be useful for noisy data (where $Bf \neq c$)?" Our architecture is designed for exactly this scenario via a two-stage process:
>
> **Stage 1 (Sec 3.1-3.3): Candidate Generation:** First, our model finds the best candidate:  $f^{{cand}}\_{\theta} = f^{(0)} + U\alpha\_\theta$. This candidate is forced to live on the perfectly balanced manifold ($Bf^{{cand}}_{\theta}=c$). It is our model's best, feature-driven guess of what the flow would be in an ideal, noise-free world.
>
> **Stage 2 (Sec 3.4): Noise-Handling Refinement.** Second, we treat this $f^{{cand}}\_{\theta}$ as a strong prior. The Tikhonov refinement (Eq. 9) then solves for the final flow $\tilde{f}\_{\theta}$. This step explicitly handles noise by finding a flow that is **(a)** close to the noisy data's empirical imbalance $\hat{c}$ and **(b)** close to our "ideal" GNN-based candidate $\delta^{\text{attn}}_{\theta}$.
>
> In short, the null-space is not used to model the noisy solution directly. Instead, it parameterizes the structural prior, the manifold where the flow would exist if measurements were perfect. The Tikhonov refinement then relaxes this constraint, allowing the model to deviate from the exact null-space just enough to fit the noisy observations.
>
> **W2: The Role of the GATv2 and Attention**
>
> The reviewer’s understanding is correct. The GATv2 is the "brain" that learns the coefficients $\\alpha$ for the null-space basis $U$ based on the edge features $X$. The GATv2 (Sec 3.3) is the core of our "learning" component. It maps the graph structure and edge features $X$ to the latent coefficients $\alpha_{\\theta}$. This is what allows our model to make a data-driven choice from the "menu" of possible physics-preserving adjustments provided by the basis $U$.
>
> **W3: The Tikhonov Refinement and Training**
>
> The reviewer’s understanding is again correct. The Tikhonov step (Sec 3.4) is not an iterative optimizer. It is a strictly convex problem with a single, closed-form solution. As the reviewer notes, we use implicit differentiation (Sec 3.5) to compute the exact gradient by solving one adjoint linear system. This is stable, memory-efficient, and avoids unrolling any iterations.

---

> ### Author Response · Authors · 2025-11-21
> **Response to Reviewer ietJ 2/2**
>
> **W4 - Overstatement and "Unnecessary" Complexity**
>
> The reviewer's feedback made us realize our complex language was a barrier. We have revised the abstract, introduction, and discussion to use the clearer, more direct language from this rebuttal.
>
> To directly test if our method's "complexity" is justified, we implemented a simpler "Predict-then-Project" (PnP) baseline (as suggested by Reviewer yn4E). This simpler model first uses a GNN to predict an unconstrained flow correction $z$, and then projects it onto the physical subspace: $\\Delta = P_{\\mathcal{A}} z$.
> The results show this "simpler" approach performs significantly worse than our method and, critically, worse than a simple MLP that ignores physics entirely. The reason is, it is far more effective to learn directly in the $k$-dimensional latent space of valid corrections (as FlowSymm does) than to learn an unconstrained signal and hope that a projection onto that space yields a good result. Our method provides a more direct, stable, and efficient learning signal, which is the key to its performance.
>
> | Model | Traffic (RMSE) | Power (RMSE) | Bike (RMSE) |
> | :--- | :---: | :---: | :---: |
> | MLP (No Physics) | 0.083 | 0.083 | 0.073 |
> | PnP Baseline (Simple Physics) | 0.201 | 0.127 | 0.127 |
> | **FlowSymm (Ours)** | **0.057** | **0.026** | **0.029** |
> *Table 1: New Experiment: FlowSymm vs. "Predict-then-Project" (PnP) Baseline. The PnP model's poor performance justifies our method's design.*
>
> **Questions**
>
> We thank the reviewer for this detailed list.
>
> **Line 106: “while respecting these physical constraints”** Yes, this refers to the flow conservation constraints $Bf=c$. We have clarified this in the text.
>
> **Section 3.3: Intuition for how GATv2 couples features to recovery**
> A basis vector $u^{(i)}$ is a global flow redistribution (e.g., a flow cycle). A GNN is needed to decide which of these global actions is appropriate. First, the GATv2 computes a feature embedding $H_e$ for each edge $e$ that is aware of its local neighborhood. Each basis vector technically represents a global adjustment that spans multiple edges. Next, we compute a "compatibility score" $q \_{e,i} = (w_i^\top H_e) |u^{(i)} \_e|$ (Eq. 7). This score is a "vote" from the local context of edge $e$ in favor of (or against) the global action $u^{(i)}$. Finally, we sum these votes from all missing edges to get a global score $s_i$, which becomes our final attention $\\alpha_{\\theta}$. This mechanism allows local, feature-rich information (e.g., "this is a 3-lane highway next to an off-ramp") to influence the selection of global, physics-preserving corrections.
>
> **Line 284: “SPD solver”** This is a typo. We have corrected it to "SPD solve" in the revised paper. Thank you for catching it.
>
> **Section 4: Edge features for each problem**
> For traffic dataset: the latitude/longitude of its upstream and downstream endpoints; the number of lanes; the posted speed limit; a hot encoding of highway type (e.g., motorway, motorway link, trunk); and three graph-based statistics such as the in-degree and PageRank centrality of the source vertex,
> and the out-degree of the target vertex.
>
> For power dataset:  reactance $\(x\)$, resistance $\(r\)$, nominal capacity $\(s\_{nom}\)$, a binary flag indicating whether $\(s_{\mathrm{nom}}\)$ is extendable, the capital cost of capacity expansion, line length, number of parallel circuits, and optimized capacity $\(s_{{nom}}^{{opt}}\)$; for bus edges the control strategy (PQ, PV, or slack) via one-hot encoding.
>
> For bike dataset: latitude/longitude of both locations, duration of trip, rideable type (classic / electric), fraction of trips by members, fraction of trips in peak hours (5-7PM and 7-9AM), and distance between locations.
>
> **Sections 4.1, 4.2: “ten baselines” or “eight baselines”?**
> This was a typo. Our Table 1 correctly lists and evaluates against nine models. We have corrected the text to "nine baselines" throughout.
>
> **Line 412: “imptovements”**
> Corrected to "improvements". Thank you.
>
> We hope these clarifications, the new experimental results, and the extensive revisions to the paper's text fully resolve the reviewer's concerns about presentation and soundness. We are confident that the revised paper is substantially clearer and more accessible. We thank the reviewer again for their time and constructive criticism. We are happy to provide any further clarifications as needed.

---

### Official Review · Reviewer_wnzU · 2025-11-01

**Soundness:** 3
**Presentation:** 3
**Contribution:** 3
**Rating:** 6
**Confidence:** 3

**Summary:**

This paper introduces FlowSymm, a solver that augments a graph attention backbone with symmetry-preserving, physics-aware corrections for partially observed flow graphs. This method for estimating missing flows in networks (e.g., traffic, power, bike-sharing) where only a subset of edges have sensors.

Its main contributions are:
1、It reinterprets admissible divergence-free adjustments as elements of an Abelian group action.
2、It combines this basis with edge-wise GATv2 embeddings. An attention mechanism scores each basis vector in a context-aware manner, enabling the model to inject corrective flows where the local structure demands, while preserving flow balance.
3、It imposes soft physical consistency through a feature-conditioned Tikhonov refinement and trains all weights end-to-end via reverse-mode implicit differentiation within a bilevel optimization framework.
4、Extensive experiments on three real-world benchmarks show that FlowSymm consistently improves upon nine baselines, reducing RMSE by up to ten percent and yielding performance gains over the previous state-of-the-art.

**Strengths:**

This is a very well-written and clearly presented paper. The core idea—leveraging a group-action basis for divergence-free corrections in network flow completion. The methodology is explained with remarkable clarity, and the experimental results are thorough and convincing. The core innovation is the reformulation of the flow completion problem through the lens of group theory. While GNNs typically encode spatial symmetries or permutation symmetries, this work defines a new, problem-specific "algebraic symmetry" derived directly from the graph's incidence matrix and the sensor mask.

In addition, the mathematical derivation is sound, from the initial balanced anchor using the Moore-Penrose pseudoinverse to the construction of the projector P_A and the implicit differentiation for the bilevel problem. The method is built on a firm algebraic foundation.

Finally, the experimental design is robust. The use of three distinct, real-world domains (Traffic, Power, Bike) demonstrates generalizability.

**Weaknesses:**

1、The introduction does an excellent job of covering related work in physics-informed ML and equivariant GNNs. However, the transition to the specific "algebraic symmetry" of flow conservation could be slightly more explicit for a reader unfamiliar with this background.
2、The derivation of the projector P_A is technically correct but may be dense for some readers. The step of subtracting the projector onto the observed edges within the balanced space is crucial but could be briefly motivated with an intuitive phrase.

**Questions:**

Questions：The entire method, starting with the "balanced anchor"  , assumes the net nodal injection vector c is known. In many real-world scenarios (e.g., unmonitored power grid feeders, unobserved traffic sources/sinks), c is also partially unknown or highly uncertain. How sensitive is FlowSymm's performance to errors in c?

**Details Of Ethics Concerns:**

Nothing

---

> ### Author Response · Authors · 2025-11-21
> **Response to Reviewer wnzU 1/2**
>
> **We sincerely thank reviewer** wnzU for their detailed, positive, and insightful review. We are greatly encouraged that the reviewer found our paper very well-written and clearly presented, our core idea a "core innovation," the mathematical derivation sound, and the experimental design robust.
>
> We are especially grateful that the reviewer recognized the novelty of our problem-specific algebraic symmetry, which distinguishes our work from typical GNNs that focus on spatial or permutation symmetries. The weaknesses identified are important points for clarification, which we have addressed by revising the paper. We have also conducted a new, detailed sensitivity analysis to definitively answer the reviewer's question about the injection vector $c$. We believe these updates, which will be incorporated into the final paper, fully address all concerns.
>
> **W1. Clarification on the transition to "algebraic symmetry"**
>
> We have revised the Introduction (Sec 1.1) to be more explicit about the distinction between our **algebraic symmetry** and traditional **spatial/permutation** symmetries. We fully agree with the reviewer that this distinction is crucial. As the reviewer noted, most equivariant GNNs handle spatial (e.g., Euclidean, E(n)) or permutation symmetries. These symmetries relate to the graph's embedding in space or the labeling of its nodes.
>
> Our work defines a constraint-based symmetry. Our **group action**, $u \in \mathcal{A}$, defines a set of adjustments that preserve the core physics and known data of the problem. Any valid adjustment $u$ must satisfy two constraints simultaneously:
>
> **Physics Preservation ($Bu=0$):** The adjustment must be divergence-free. Adding it to a balanced flow $f$ must result in a new flow $f' = f+u$ that is also balanced: $Bf' = B(f+u) = Bf + Bu = c + 0 = c$.
>
> **Sensor Invariance ($S_{\text{obs}}u=0$):** The adjustment must not alter any of the known sensor readings on the observed edges $\mathcal{E}_{\text{obs}}$.
>
> We have added text to **Section 1.1** to explicitly state that while E(n)-GNNs encode spatial symmetries, our architecture is designed to respect this algebraic, constraint-preserving symmetry, which is defined by the graph's incidence matrix ($B$) and the sensor mask ($S_{\text{obs}}$).
>
> **W2. Intuitive motivation for the $P_{\mathcal{A}}$ projector**
>
> We have added a clarifying sentence in Section 3.2 to provide a more intuitive, step-by-step motivation for the $P_{\mathcal{A}}$ projector's formula. The reviewer is correct that the formula for $P_{\mathcal{A}}$ in Section 3.2 is dense. We have added text to explain its derivation more intuitively.
> The formula: $$P_{\mathcal{A}} = P_{\text{bal}} - P_{\text{bal}} S_{\text{obs}}^{\top} (S_{\text{obs}} P_{\text{bal}} S_{\text{obs}}^{\top})^{\dagger} S_{\text{obs}} P_{\text{bal}}$$
> can be understood as a two-step "filtering" process:
>
> **Step 1:** We start with the projector for all divergence-free flows, $P_{\text{bal}} = I - B^{\dagger}B$, which projects any vector onto $\ker B$.
>
> **Step 2:** From this set, we remove any components that would "leak" onto the observed edges and change their values. The large term being subtracted is a standard projection formula that finds and removes all components of $P_{\text{bal}}$ that are not zero on $\mathcal{E}_{\text{obs}}$.
>
> We have updated **Section 3.2** to explain this. The new text clarifies that **"Intuitively, this formula first takes the projector onto all balanced flows ($P_{\text{bal}}$) and then explicitly subtracts the projection onto the 'invalid' subspace of balanced flows that would alter the sensor readings. The result, $P_{\mathcal{A}}$, is the projector onto the exact subspace of valid, physics-preserving, and sensor-respecting adjustments.**"

---

> > ### Author Response · Authors · 2025-11-21
> > **Response to Reviewer wnzU 2/2**
> >
> > **Q1. Sensitivity of FlowSymm to errors in the injection vector $c$**
> >
> > The reviewer's question is critical and practical, asking about the effect of noisy $c$. We checked the effect on: (i) the anchor $f^{(0)}$, (ii) the subspace/basis $U$, and (iii) the final predictions $\tilde{f}_{\theta}$. Our new experiment, which trains on the true $c$ and evaluates on a corrupted $c' = c + \epsilon$, allows us to answer each part definitively.
> >
> > **- (i) Effect on Anchor ($f^{(0)}$):** The anchor $f^{(0)}$ is directly and mechanically sensitive to $c$. As shown in our paper in Section 3.1, the anchor's missing components $\delta^{(0)}$ are computed directly from $c$: $\delta^{(0)} = B_{miss}^{\dagger} (c - B_{\text{obs}}\hat{f}^{\text{obs}})$.
> > -   Our experiment confirms this. As shown in the table below, the L2-norm of the anchor component (Anchor Norm) visibly drifts as noise is added. For example, on the Traffic dataset, the anchor norm drifts from 17.44 (at 0% noise) to 18.44 (at 50% noise).
> >
> > **-   (ii) Effect on Subspace/Basis ($\mathcal{A}, U$):**
> >     -   The subspace $\mathcal{A}$ and its basis $U$ are completely independent of $c$. As defined in Section 3.2, the admissible subspace is $\mathcal{A} = \{ u \in \mathbb{R}^m \mid Bu = 0 \text{ and } S_{\text{obs}}u = 0 \}$.
> > -   This definition depends only on the graph topology ($B$) and the sensor mask ($S_{\text{obs}}$), not on the injection vector $c$. Therefore, noise in $c$ has zero effect on the basis $U$ that spans this space.
> >
> > **-   (iii) Effect on Final Predictions ($\tilde{f}_{\theta}$):**
> >  -   The final predictions are highly robust, which is a key strength of our model.
> > -   Our model's final candidate $\delta^{\text{attn}}_{\theta}$ (before refinement) is the sum of the $c$-dependent anchor $f^{(0)}$ and the GNN-guided correction $\Delta = U\alpha$.
> > -   Our experiment shows that while the Anchor Norm (from $f^{(0)}$) drifts with noise, the Action Norm(from $\Delta$) is constant. This is because $\Delta$ is learned from edge features, which are $c$-independent.
> > -   As the table shows, the final Test RMSE remains remarkably stable, degrading only from 0.057 to 0.060 on Traffic, even with 50% noise in $c$. This demonstrates that FlowSymm is not merely a refinement of a naive anchor. It learns a powerful, feature-based, and physics-preserving correction $\Delta$ that is independent of $c$ and robustly dominates the (potentially noisy) $c$-dependent anchor.
> >
> > **Table: FlowSymm Sensitivity to Noise in Injection Vector \(c\)**
> > We report the L2-norm of the \(c\)-dependent anchor $f^{(0)}$, the L2-norm of the learned GNN correction  $\Delta$, and the final Test RMSE as \(c\)-noise increases.
> >
> > | **Dataset** | **Noise Level** | **Anchor Norm** | **Action Norm** | **Test RMSE** |
> > |-------------|------------------|--------------------------------------|-------------------------------------|----------------|
> > | **Traffic** | 0% Noise         | 17.4456 | 5.3303 | **0.057** |
> > |             | 15% Noise        | 17.5387 | 5.3303 | **0.057** |
> > |             | 35% Noise        | 18.2090 | 5.3303 | **0.058** |
> > |             | 50% Noise        | 18.4457 | 5.3303 | **0.060** |
> > | **Power**   | 0% Noise         | 2.1851 | 1.3777 | **0.026** |
> > |             | 15% Noise        | 2.1859 | 1.3777 | **0.026** |
> > |             | 35% Noise        | 2.2047 | 1.3777 | **0.027** |
> > |             | 50% Noise        | 2.2147 | 1.3777 | **0.028** |
> > | **Bike**    | 0% Noise         | 2.2736 | 1.0450 | **0.029** |
> > |             | 15% Noise        | 2.2680 | 1.0450 | **0.029** |
> > |             | 35% Noise        | 2.2642 | 1.0450 | **0.029** |
> > |             | 50% Noise        | 2.3169 | 1.0450 | **0.030** |
> >
> > We have added these clarifications and the new sensitivity analysis to the Appendix.
> > We thank the reviewer again for their constructive feedback and are happy to provide any further clarifications as needed.

---

### Author Response · Authors · 2025-11-21
**Summary of rebuttal**

We sincerely thank you for your time, thorough evaluation, and constructive feedback. We are encouraged by the consensus that our problem is important, our method is novel, and our results show strong potential.

We have completed our revisions and uploaded an updated manuscript. Your comments have helped us improve both the clarity of our presentation and the empirical validation of our approach. We summarize the key updates and the additional experiments:

**1. Clarifications on Terminology and Intuition**

- Based on feedback regarding presentation (Reviewers ietj, wnzU), we have revised the Abstract, Introduction, and Method sections:
We explicitly clarified the distinction between our constraint-based "algebraic symmetry" (defined by conservation laws and sensor masks) and the spatial/geometric symmetries in traditional E(n)-equivariant GNNs.

- We have clarified the role of the Abelian group structure while retaining the original terminology, and we now relate it more explicitly to concepts such as the null-space basis and solution subspace to better align with readers’ intuition about linear systems.

- We added a step-by-step, intuitive explanation for the construction of the projector $P_\\mathcal{A}$, describing it as a process of filtering out components of the balanced flow that would violate sensor constraints.

**2. New Experimental Results (Appendix E)**

- We conducted three new sets of experiments to directly address questions regarding robustness and baselines. These are now detailed in the new Appendix E:
    - **Sensitivity to Injection Vector $c$** (Reviewers wnzU, yn4E): We performed a sensitivity analysis by training on ground-truth $c$ and injecting noise at test time. While the anchor component $f^{(0)}$ drifts with noise as expected, our model's learned, feature-based correction ($\\Delta$) remains stable. Consequently, the final RMSE remains robust (e.g., increasing only from 0.057 to 0.060 on Traffic even with 50% noise).
    - **"Predict-then-Project" (PnP) Baseline (Reviewer yn4E)**: We implemented the suggested PnP baseline, which predicts an unconstrained vector and projects it onto the feasible subspace. FlowSymm significantly outperforms this baseline (e.g., 0.02 vs. 0.13 RMSE on Power), and notably, the PnP baseline performs worse than a simple physics-agnostic MLP. This validates our architectural choice to learn coefficients within the latent basis rather than projecting onto it.
   - **Physical Consistency Analysis (Reviewers MoiE, yn4E)**: We computed the final divergence residuals ($||B\tilde{f} - c||_2$) for all models.
Result: FlowSymm achieves the lowest divergence residual across all datasets while maintaining the lowest RMSE, proving that it achieves the best balance of data fidelity and physical adherence.

We have posted detailed, individual responses to each reviewer addressing their specific questions. We believe these revisions and additional analyses fully address the concerns. We remain open to further discussion and are happy to provide any additional clarifications during the discussion period.

Best regards,

The Authors

---

### Comment · Area_Chair_cQzi · 2025-11-27
**Reviewers, please respond**

Dear reviewers,

please respond to the authors so that they have time before the discussion period ends.

Thanks
Your AC

---

### Author Response · Authors · 2025-11-30
**Message to the new area chair**

**To the new Area Chair:**

We understand the unique circumstances regarding the re-assignment of this paper and the freezing of reviewer scores. We appreciate your time in evaluating our work under these constraints.

Given that the original reviewers are currently unable to update their scores or continue the discussion to reflect our rebuttal, we provide this summary to assist you. We have uploaded a revised manuscript which includes three critical experiments requested during the review process.

The main topics during the rebuttal process were as follows:

### **1. Robustness to Noise in Injection Vector  c**

-   **The Concern (Reviewers wnzU, yn4E):**  Since the initial anchor  $f^{(0)}$  depends on the injection vector $c$, reviewers asked if measurement errors or uncertainty in $c$ would destabilize the model's predictions.

-   **Our New Evidence (Appendix E.1):**  We performed a rigorous sensitivity analysis, training the model on ground-truth  $c$ and injecting Gaussian noise (up to 50%) at test time.

-   **The Result:**  The experiment confirms that while the physics-based anchor drifts with noise, our model's  **learned, feature-based correction remains stable**. Consequently, the final Test RMSE is exceptionally robust, increasing negligibly (e.g., from  **0.057 to 0.060**  on Traffic) even when $c$ is corrupted with  **50% noise**.

### **2. Justification of Latent Learning vs. Simple Projection**

-   **The Concern (Reviewer yn4E):**  Is it necessary to learn coefficients  _within_  the basis, or could a simpler "Predict-then-Project" (PnP) baseline work?

-   **Our New Evidence (Appendix E.2):**  We implemented the suggested PnP baseline, which trains an identical GNN to predict an unconstrained vector and projects it onto the feasible subspace.

-   **The Result:**  FlowSymm drastically outperforms PnP (e.g.,  **0.026 vs. 0.127 RMSE**  on Power). Crucially, the  **PnP baseline performs worse than a simple physics-agnostic MLP**, proving that the projection step destroys gradient signals in naive approaches. This decisively validates our architectural choice to learn  _within_  the valid subspace.


### **3. Verification of Physical Consistency**

-   **The Concern (Reviewer MoiE, yn4E):**  Does the model actually respect conservation laws better than physics-agnostic baselines?

-   **Our New Evidence (Appendix E.3):**  We computed the final Divergence Residual  ($||B\tilde{f} - c||_2$) for all models.

-   **The Result:**  FlowSymm achieves the  **lowest divergence residual**  across all datasets compared to physics-agnostic baselines. This confirms our method achieves the optimal balance of high data fidelity and strict physical adherence.

### **4. Presentation and Accessibility**

-   **The Concern (Reviewer ietj):**  Some mathematical terminology (e.g., "Abelian group," "algebraic symmetry") was found to be dense for non-experts.

-   **Our Revision:**  We have revised the  **Abstract, Introduction, and Method**  sections. We now explicitly clarified the role of the Abelian group structure while retaining the original terminology, and we now relate it more explicitly to concepts such as the null-space basis and solution subspace to better align with readers’ intuition about linear systems. We have added a step-by-step intuitive explanation for the construction of the projector $P_\\mathcal{A}$.

We believe these new experimental results definitively resolve the technical questions raised. We are confident the revised manuscript presents a robust, interpretable, and state-of-the-art solution for network flow completion. While the reviewers cannot currently respond, we remain fully available to you during the remainder of the discussion period should you have any specific questions or require further clarification on these new results.

Best regards,

The Authors

---

> ### Comment · Area_Chair_cZJR · 2025-12-02
>
> Quick question -- did you upload a new version of your manuscript that addressed the reviewer comments?

---

> ### Author Response · Authors · 2025-12-02
>
> Yes, we have uploaded the revised manuscript. It incorporates the requested terminology clarifications and adds a new Appendix E, which details three critical experiments (sensitivity analysis, PnP baseline, and physical consistency) that directly address the reviewers' concerns.

---

### Meta-Review · Area_Chair_cZJR · 2026-01-06

**Summary:**

**Summary:**
This paper introduces FlowSymm, a method for recovering missing edge flows in networks with conservation laws. The approach is validated on multiple different datasets, yielding between an 8-10% RMSE improvement over prior sota.

**Rationale:**
Among the 4 reviewers, there was significant disagreement with 3 reviewers recommending acceptance and one reviewer taking a hard line on rejection. I'm choosing to ignore reviewer ietj's review almost entirely as it amounts to pointing out that a paper on a mathematically dense topic uses mathematically precise language and notation -- this feels like a mismatch between reviewer and paper which is sadly not uncommon given the explosion in submissions over the last decade. With that out of the way, the remaining three reviews all coalesced on an Accept and I'm inclined to agree.

**Reviewer Concerns:**

wnzu:
- algebraic symmetry: The authors clarified constraint based algebraic symmetric versus geometric symmetries and revised the introduction accordingly
- noise injection: The authors addressed this concern through synthetic noise injection experiments

ietj:
- Mathematically dense: Authors partially addressed this, but to my eye the paper isn't particularly dense. It uses precise terms of art in places, but nothing is out of place or there solely for mathematical flourish.
- overstatement of claims: The authors provided a valid justification for their claims, pushing back with convincing evidence

moie:
- Limited flexibility: The authors clarified that you only need a single basis to handle global consistency, and showed this through bil-gcn experiments which have per-edge regularization
- Lack of physics preserving metrics: The authors added this
- Missing comparisons to neural operators: The authors acknowledged this limitation but pointed out differences in domain (eg discrete vs continuous)

yn4e:
- basis choice: The authors clarified that the SVD-ordered basis was a conscious choice
- predict-then-project baseline: The author added this
- noise sensitivity: The authors addressed this concern through synthetic noise injection experiments
- dof overestimation: The authors acknowledged that this can occur

**Reviewer Scores:**

wnzu, 6->6, concerns about c-sensitivity were addressed but the concerns were not so large as to yield an upgrade to an 8
ietj, 2->4, Presentation issues were the primary concern, but were not shared by the rest of the reviewers or the AC. Other concerns such as additional baselines were addressed.
MoiE, 6->6, all concerns were addressed, but the concerns were not so large as to yield an upgrade to an 8
yn4e, 6->6, all questions were addressed experimentally.

---

### Decision · Program_Chairs · 2026-01-26

Accept (Poster)